# Act As You Wish: Fine-Grained Control of Motion Diffusion Model with Hierarchical Semantic Graphs

Peng Jin[1,4]  Yang Wu[3*]  Yanbo Fan[3]  Zhongqian Sun[3]  Yang Wei[3]  Li Yuan[1,2,4*]

[1] School of Electronic and Computer Engineering, Peking University, Shenzhen, China
[2] Peng Cheng Laboratory, Shenzhen, China  [3] Tencent AI Lab, China
[4] AI for Science (AI4S)-Preferred Program, Peking University Shenzhen Graduate School, China

jp21@stu.pku.edu.cn  dylan.yangwu@qq.com  yuanli-ece@pku.edu.cn

## Abstract

Most text-driven human motion generation methods employ sequential modeling approaches, e.g., transformer, to extract sentence-level text representations automatically and implicitly for human motion synthesis. However, these compact text representations may overemphasize the action names at the expense of other important properties and lack fine-grained details to guide the synthesis of subtly distinct motion. In this paper, we propose hierarchical semantic graphs for fine-grained control over motion generation. Specifically, we disentangle motion descriptions into hierarchical semantic graphs including three levels of motions, actions, and specifics. Such global-to-local structures facilitate a comprehensive understanding of motion description and fine-grained control of motion generation. Correspondingly, to leverage the coarse-to-fine topology of hierarchical semantic graphs, we decompose the text-to-motion diffusion process into three semantic levels, which correspond to capturing the overall motion, local actions, and action specifics. Extensive experiments on two benchmark human motion datasets, including HumanML3D and KIT, with superior performances, justify the efficacy of our method. More encouragingly, by modifying the edge weights of hierarchical semantic graphs, our method can continuously refine the generated motion, which may have a far-reaching impact on the community. Code and pre-trained weights are available at https://github.com/jpthu17/GraphMotion.

## 1 Introduction

Human motion generation is a fundamental task in computer animation [54, 4] and has many practical applications across various industries including gaming, film production, virtual reality, and robotics [61, 8]. With the progress made in recent years, text-driven human motion generation has allowed for the synthesis of a variety of human motion sequences based on natural language descriptions. Therefore, there is a growing interest in generating manipulable, plausible, diverse, and realistic sequences of human motion from flexible natural language descriptions.

Existing text-to-motion generation methods [42, 61, 8, 1, 63] mainly rely on sentence-level representations of texts and directly learn the mapping from the high-level language space to the motion sequences. Recently, some works [54, 8, 62] propose conditional diffusion models for human motion synthesis and further improve the synthesized quality and diversity. Although these methods have made encouraging progress, they are still deficient in the following two aspects. (i) **Imbalance.** The model, which directly uses the transformers [55] to extract text features automatically and implicitly, may overemphasize the action names at the expense of other important properties like direction and

---

*Corresponding author: Yang Wu, Li Yuan.

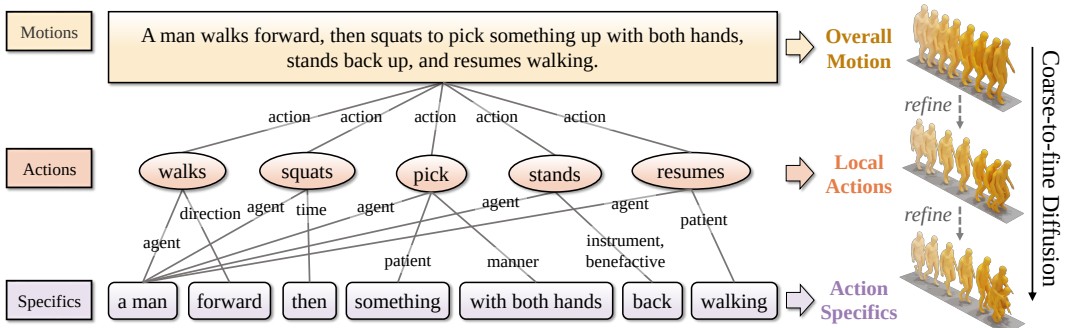

Figure 1: **We propose hierarchical semantic graphs, a fine-grained control signal, for text-to-motion generation and factorize text-to-motion generation into hierarchical levels including motions, actions, and specifics to form a coarse-to-fine structure.** This approach enhances the fine-grained correspondence between textual data and motion sequences and achieves better controllability conditioning on hierarchical semantic graphs than carefully designed baselines.

intensity. As a typical consequence of this unbalanced learning, the network is insensitive to the subtle changes in the input text and lacks fine-grained controllability. (ii) **Coarseness.** On the one hand, motion descriptions frequently refer to multiple actions and attributes. However, the compact sentence-level representations extracted by current works usually fail to convey the clarity and detail needed to fully understand the text, leading to a lack of fine-grained details to guide the synthesis of subtly distinct motion. On the other hand, mapping directly of existing works from the high-level language space to motion sequences further hinders the generation of fine-grained details. Therefore, we argue that it is time to seek a more precise and detailed text-driven human motion generation method to ensure an accurate synthesis of complex human motions.

To this end, we propose a more fine-grained control signal, hierarchical semantic graphs, to represent different intentions for controllable motion generation and design a coarse-to-fine motion diffusion model, called GraphMotion. As shown in Fig. 1, motion descriptions inherently possess hierarchical structures and can be represented as hierarchical graphs composed of three types of abstract nodes, namely motions, actions, and specifics. Concretely, the overall sentence describes the global motion involving multiple actions, e.g., "walk", "pick", and "stand" in Fig. 1, which occur in sequential order. Each action consists of different specifics that act as its attributes, such as the agent and patient of the action. Such global-to-local structures contribute to a reliable and comprehensive understanding of motion descriptions. Correspondingly, to take full advantage of this fine-grained control signal, we decompose the text-to-motion diffusion process into three semantic levels from coarse to fine, which are responsible for capturing the overall motion, local actions, and action specifics, respectively.

The proposed GraphMotion has three compelling advantages: **First**, the explicit factorization of the language embedding space enables us to build a fine-grained correspondence between textual data and motion sequences, which avoids the imbalanced learning of different textual components and coarse-grained control signal representation. **Second**, the hierarchical refinement property of GraphMotion allows the model to progressively enhance the generated results from coarse to fine, which avoids the coarse-grained generated results. **Third**, to further fine-tune the generated results for more fine-grained control, our method can continuously refine the generated motion by modifying the edge weights of the hierarchical semantic graph. Experimental results on two benchmark datasets for text-to-motion generation, including HumanML3D [14] and KIT [43], demonstrate the advantages of GraphMotion. The main contributions of this work are as follows:

- To the best of our knowledge, we are the first to propose hierarchical semantic graphs, a fine-grained control signal, for text-to-motion generation. It decomposes motion descriptions into global-to-local three types of abstract nodes, namely motions, actions, and specifics.

- Correspondingly, we decompose the text-to-motion diffusion process into three semantic levels. This allows the model to gradually refine results from coarse to fine. Experiments show that our method achieves new state-of-the-art results on two text-to-motion datasets.

- More encouragingly, by modifying the edge weights of hierarchical semantic graphs, our method can continuously refine the generated motion, which may have a far-reaching impact.

## 2 Related Work

**Text-driven Human Motion Generation.** Text-driven human motion generation aims to generate 3D human motion based on text descriptions. Due to the user-friendliness and convenience of natural language [22], text-driven human motion generation is gaining significant attention and has many applications. Recently, two categories of motion generation methods have emerged: joint-latent models [2, 42] and diffusion models [8, 62, 47]. Joint-latent models, e.g., TEMOS [42], typically learn a motion variational autoencoder and a text variational autoencoder. These models then constrain the text and motion encoders into a shared latent space using the Kullback-Leibler divergences [29] loss. The latter category of methods, e.g., MDM [54], proposes a conditional diffusion model for human motion generation to learn a powerful probabilistic mapping from the textual descriptors to human motion sequences. Although these methods have made encouraging progress, they still suffer from two major deficiencies: unbalanced text learning and coarse-grained generated results. In this paper, we propose to leverage the inherent structure of language [6, 58] for motion generation. Specifically, we introduce hierarchical semantic graphs as a more effective control signal for representing different intentions and design a coarse-to-fine motion diffusion model using this signal.

**Diffusion Generative Models.** Diffusion generative models [49, 18, 11, 21, 50] are a type of neural generative model that uses the stochastic diffusion process, which is based on thermodynamics. The process involves gradually adding noise to a sample from the data distribution, and then training a neural network to reverse this process by gradually removing the noise. In recent years, diffusion models have shown promise in a variety of tasks such as image generation [18, 50, 11, 19, 57], natural language generation [3, 36, 13], as well as visual tasks [9]. Some other works [25, 32] have attempted to adapt diffusion models for cross-modal retrieval [33, 34, 35, 23, 24]. Inspired by the success of diffusion generative models, some works [62, 54, 8] have applied diffusion models to human motion generation. However, these methods typically learn a one-stage mapping from the high-level language space to motion sequences, which hinders the generation of fine-grained details. In this paper, we decompose the text-to-motion diffusion process into three semantic levels from coarse to fine. The resultant levels are responsible for capturing overall motion, local actions, and action specifics, which enhances the generated results progressively from coarse to fine.

**Graph-based Reasoning.** The graph convolutional network [28] is originally proposed to recognize graph data. It uses convolution on the neighborhoods of each node to produce outputs. Graph attention networks [56] further enhance graph-based reasoning by dynamically attending to the features of neighborhoods. The graph-based reasoning has shown great potential in many tasks, such as scene graph generation [60], visual question answering [20, 31, 30], natural language generation [6], and cross-modal retrieval [7]. In this paper, we focus on reasoning over hierarchical semantic graphs on motion descriptions for fine-grained control of human motion generation.

## 3 Methodology

In this paper, we tackle the tasks of text-driven human motion generation. Concretely, given an arbitrary motion description, our goal is to synthesize a human motion $x^{1:L} = \{x^i\}_{i=1}^{L}$ of length $L$. The overview of the proposed GraphMotion is shown in Fig. 2.

### 3.1 Hierarchical Semantic Graph Modeling

Existing works for text-driven human motion generation typically directly use the transformer [55] to extract text features automatically and implicitly. However, motion descriptions inherently possess hierarchical structures that can be divided into three sorts of abstract nodes, including motions, actions, and specifics. Compared with the sequential structure, such global-to-local structure contributes to a reliable and comprehensive understanding of the semantic meaning of motion descriptions and is a promising fine-grained control signal for text-to-motion generation.

**Semantic Role Parsing.** To obtain actions, attributes of action as well as the semantic role of each attribute to the corresponding action, we implement a semantic parser of motion descriptions based on a semantic role parsing toolkit [48, 7]. We extract three types (motions, actions, and specifics) of nodes and twelve types of edges to represent various associations among the nodes. For details about the types of nodes and edges, please refer to our supplementary material.

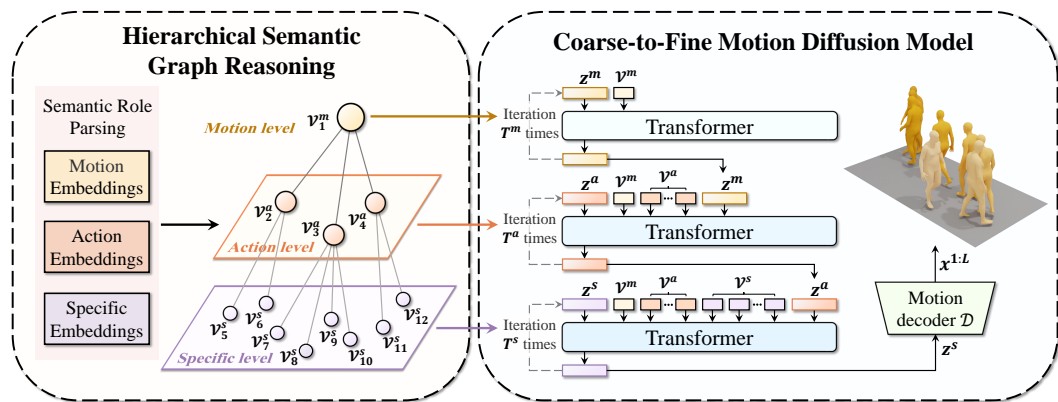

Figure 2: **The overview of the proposed GraphMotion for text-driven human motion generation.** We factorize motion descriptions into hierarchical semantic graphs including three levels of motions, actions, and specifics. Correspondingly, we decompose the text-to-motion diffusion process into three semantic levels, which correspond to capturing the overall motion, local actions, and action specifics.

Specifically, given the motion description, the parser extracts verbs that appeared in the sentence and attribute phrases corresponding verb, and the semantic role of each attribute phrase. The overall sentence is treated as the global motion node in the hierarchical graph. The verbs are considered as action nodes and connected to the motion node with direct edges, allowing for implicit learning of the temporal relationships among various actions during graph reasoning. The attribute phrases are specific nodes that are connected with action nodes. The edge type between action and specific nodes is determined by the semantic role of the specifics in relation to the action.

**Graph Node Representation.** Given the motion description, we follow previous works [53, 54, 61, 8] and leverage the text encoder of CLIP [45] to extract the text embedding. For the global motion node $v^m$, we utilize the [CLS] token to summarize the salient event described in the sentence. For the action node $v^a$, we use the token of the corresponding verb as the action node representation. For the specific node $v^s$, we apply mean-pooling over tokens of each word in the attribute phrase.

**Action-aware Graph Reasoning.** The interactions across different levels in the constructed graph not only explain the properties of local actions and how local actions compose the global motion, but also reduce ambiguity at each node. For example, the verb "pick" in Fig. 1 can represent different actions without context, but the context "with both hands" constrains its semantics, so that it represents the action of "pick up with both hands" rather than "pick up with one hand." Therefore, we propose to reason over interactions in the graph to obtain hierarchical textual representations.

We utilize graph attention networks [56] (GAT) to model interactions in a graph. Specifically, given the initialized node embeddings $v = \{v^m, v^a, v^s\}$, we first transform the input node embeddings into higher-level embeddings $h = \{h^m, h^a, h^s\}$ by:

$$h^m = \boldsymbol{W}v^m, \quad h^a = \boldsymbol{W}v^a, \quad h^s = \boldsymbol{W}v^s, \tag{1}$$

where $\boldsymbol{W} \in \mathbb{R}^{D \times D}$ is a shared linear transformation and $D$ is the dimension of node representation. For each pair $\{h_i, h_j\}$ of connected nodes, we concatenate the node $h_i \in \mathbb{R}^D$ with its neighbor node $h_j \in \mathbb{R}^D$, generating the input data $\tilde{h}_{ij} = [h_i, h_j] \in \mathbb{R}^{2D}$ of the graph attention module.

However, in a graph with multiple types of edges, the vanilla graph networks need to learn separate transformation matrices for each edge type. This can be inefficient when learning from limited motion data, and prone to over-fitting on rare edge types.

To this end, we propose to factorize multi-relational weights into two parts: a common transformation matrix $\boldsymbol{M} \in \mathbb{R}^{2D \times 1}$ that is shared for all edge types and a relationship embedding matrix $\boldsymbol{M_r} \in \mathbb{R}^{2D \times N}$ that is specific for different edges, where $N$ is the number of edge types. Following GAT [56], we apply LeakyReLU [39] in the calculation of attention coefficients and set the negative input slope to 0.2. The attention coefficient $\tilde{e}_{ij}$ is formulated as:

$$e_{ij} = \text{LeakyReLU}(\boldsymbol{M}^\top \tilde{h}_{ij}) + \text{LeakyReLU}(R_{ij}\boldsymbol{M_r}^\top \tilde{h}_{ij}), \quad \tilde{e}_{ij} = \frac{\exp(e_{ij})}{\sum_{k \in \mathbb{N}_i} \exp(e_{ik})}, \tag{2}$$

where $R_{ij} \in \mathbb{R}^{1 \times N}$ is a one-hot vector denoting the type of edge between node $i$ and $j$. $\mathbb{N}_i$ is the set of neighborhood nodes of node $i$. To alleviate over-smoothing [59] in graph networks, we apply skip connection when calculating output embeddings. The output embeddings $\mathcal{V}$ are formulated as:

$$\mathcal{V}_i = \sigma\big(\sum_{j \in \mathbb{N}_i} \tilde{e}_{ij} h_j\big) + v_i, \tag{3}$$

where $\sigma$ is a nonlinear function. Following GAT [56], we use ELU [10] as the nonlinear function $\sigma$.

## 3.2 Coarse-to-Fine Motion Diffusion Model for Graphs

To leverage the coarse-to-fine topology of hierarchical semantic graphs during generation, we also decompose the text-to-motion diffusion process into three semantic levels, which correspond to capturing the overall motion, local actions, and action specifics. During the reverse denoising process, the fine-grained semantic layer generates results based on the results from the coarse-grained semantic layer. This allows for a detailed and plausible representation of the intended motion.

**Motion Representation.** Following previous works [42, 8, 61], we first encode the motion into the latent space with a motion variational autoencoder [27] and then use diffusion models to learn the mapping from hierarchical semantic graphs to the motion latent space.

Specifically, we build the motion encoder $\mathcal{E}$ and decoder $\mathcal{D}$ based on the transformer [55, 41]. For the motion encoder $\mathcal{E}$, we take $C$ learnable query tokens and motion sequence $x^{1:L} = \{x^i\}_{i=1}^L$ as inputs to generate motion latent embeddings $z \in \mathbb{R}^{C \times D'}$, where $D'$ is the dimension of latent representation. For the motion decoder $\mathcal{D}$, we take the latent embeddings $z \in \mathbb{R}^{C \times D'}$ and the motion query tokens as the input to generate a human motion sequence $x^{1:L} = \{x^i\}_{i=1}^L$ with $L$ frames.

The loss $\mathcal{L}_{\text{VAE}}$ of the motion variational autoencoder can be divided into two parts. First, we use the mean squared error (MSE) to reconstruct the original input. Second, we use the Kullback-Leibler divergences (KL) loss [29] to narrow the distance between the distribution of latent space $q(z|x)$ and the standard Gaussian distribution $\mathcal{N}(0, \text{I})$. The full loss $\mathcal{L}_{\text{VAE}}$ is formulated as:

$$\mathcal{L}_{\text{VAE}} = \mathbb{E}_{x \sim q(x)}\Big[ \underbrace{\|x - \mathcal{D}(\mathcal{E}(x))\|_2^2}_{\text{MSE}} + \lambda \underbrace{\text{KL}\big(\mathcal{N}(0, \text{I}) \| q(z|x)\big)}_{\text{KL}} \Big], \tag{4}$$

where $\lambda$ is the trade-off hyper-parameter. $q(z|x) = \mathcal{N}(\mu_z, \Sigma_z)$ is obtained by sampling based on the mean $\mu_z$ and variance $\Sigma_z$ estimated by the model. To generate motion from coarse to fine step by step, we encode motion independently into three latent representation spaces $z^m \in \mathbb{R}^{C^m \times D'}$, $z^a \in \mathbb{R}^{C^a \times D'}$ and $z^s \in \mathbb{R}^{C^s \times D'}$, where the number of tokens gradually increases, i.e., $C^m \leq C^a \leq C^s$.

**Hierarchical Graph-to-Motion Diffusion.** Corresponding to the three-level structure of the hierarchical semantic graphs, we decompose the diffusion process into three semantic levels and build three transformer-based denoising models, which correspond to motions, actions, and specifics.

For the motion level model $\phi_m$, our goal is to learn the diffusion process from global motion node $\mathcal{V}^m$ to motion latent representation $z^m$. In a forward diffusion process $q(z_t^m | z_{t-1}^m)$, noised sampled from Gaussian distribution is added to a ground truth data distribution $z_0^m$ at every noise level $t$:

$$q(z_t^m | z_{t-1}^m) = \mathcal{N}(z_t^m; \sqrt{1 - \beta_t} z_{t-1}^m, \beta_t \text{I}), \quad q(z_{1:T}^m | z_0^m) = \prod_{t=1}^T q(z_t^m | z_{t-1}^m), \tag{5}$$

where $\beta_t$ is the step size which gradually increases. $T$ is the length of the Markov chain. We sample $z_t^m$ by $z_t^m = \sqrt{\bar{\alpha}_t} z_0^m + \sqrt{1 - \bar{\alpha}_t} \epsilon^m$, where $\bar{\alpha}_t = \prod_{i=1}^t (1 - \beta_i)$. $\epsilon^m$ is a noise sampled from $\mathcal{N}(0, 1)$. We follow previous works [18, 8] and predict the noise component $\epsilon^m$, i.e., $\widehat{\epsilon^m} = \phi_m(z^m, t^m, \mathcal{V}^m)$.

For the action level model $\phi_a$, to leverage the results generated by the motion level, we concatenate the action node $\mathcal{V}^a$, the motion node $\mathcal{V}^m$, and the result $z^m$ generated by the motion level together as the input of the action level denoising network, i.e., $\widehat{\epsilon^a} = \phi_a(z^a, t^a, [\mathcal{V}^m, \mathcal{V}^a, z^m])$.

For the specific level model $\phi_s$, we leverage the results generated by the action level and nodes at all semantic levels to predict the noise component, i.e., $\widehat{\epsilon^s} = \phi_s(z^s, t^s, [\mathcal{V}^m, \mathcal{V}^a, \mathcal{V}^s, z^a])$.

Table 1: **Comparisons to current state-of-the-art methods on the HumanML3D test set.** "↑" denotes that higher is better. "↓" denotes that lower is better. "→" denotes that results are better if the metric is closer to the real motion. We repeat all the evaluations 20 times and report the average with a 95% confidence interval. **Bold** and underlined indicate the best and second-best results, respectively. For fair comparisons, we report results with total diffusion steps $T^m + T^a + T^s$ of 50 and 150.

| Methods | R-Precision ↑ | | | FID ↓ | MM-Dist ↓ | Diversity → | MModality ↑ |
|---|---|---|---|---|---|---|---|
| | Top-1 | Top-2 | Top-3 | | | | |
| Real motion | $0.511^{\pm.003}$ | $0.703^{\pm.003}$ | $0.797^{\pm.002}$ | $0.002^{\pm.000}$ | $2.974^{\pm.008}$ | $9.503^{\pm.065}$ | - |
| Text2Gesture [5] | $0.165^{\pm.001}$ | $0.267^{\pm.002}$ | $0.345^{\pm.002}$ | $7.664^{\pm.030}$ | $6.030^{\pm.008}$ | $6.409^{\pm.071}$ | - |
| Seq2Seq [44] | $0.180^{\pm.002}$ | $0.300^{\pm.002}$ | $0.396^{\pm.002}$ | $11.75^{\pm.035}$ | $5.529^{\pm.007}$ | $6.223^{\pm.061}$ | - |
| Language2Pose [2] | $0.246^{\pm.001}$ | $0.387^{\pm.002}$ | $0.486^{\pm.002}$ | $11.02^{\pm.046}$ | $5.296^{\pm.008}$ | $7.676^{\pm.058}$ | - |
| Hier [12] | $0.301^{\pm.002}$ | $0.425^{\pm.002}$ | $0.552^{\pm.004}$ | $6.532^{\pm.024}$ | $5.012^{\pm.018}$ | $8.332^{\pm.042}$ | - |
| MDM [54] | $0.320^{\pm.005}$ | $0.498^{\pm.004}$ | $0.611^{\pm.007}$ | $0.544^{\pm.044}$ | $5.566^{\pm.027}$ | $\mathbf{9.559^{\pm.086}}$ | $\mathbf{2.799^{\pm.072}}$ |
| TEMOS [42] | $0.424^{\pm.002}$ | $0.612^{\pm.002}$ | $0.722^{\pm.002}$ | $3.734^{\pm.028}$ | $3.703^{\pm.008}$ | $8.973^{\pm.071}$ | $0.368^{\pm.018}$ |
| TM2T [15] | $0.424^{\pm.003}$ | $0.618^{\pm.003}$ | $0.729^{\pm.002}$ | $1.501^{\pm.017}$ | $3.467^{\pm.011}$ | $8.589^{\pm.076}$ | $2.424^{\pm.093}$ |
| T2M [14] | $0.457^{\pm.002}$ | $0.639^{\pm.003}$ | $0.740^{\pm.003}$ | $1.067^{\pm.002}$ | $3.340^{\pm.008}$ | $9.188^{\pm.002}$ | $2.090^{\pm.083}$ |
| MLD [8] | $0.481^{\pm.003}$ | $0.673^{\pm.003}$ | $0.772^{\pm.002}$ | $0.473^{\pm.013}$ | $3.196^{\pm.010}$ | $9.724^{\pm.082}$ | $2.413^{\pm.079}$ |
| MotionDiffuse [62] | $0.491^{\pm.001}$ | $0.681^{\pm.001}$ | $\underline{0.782^{\pm.001}}$ | $0.630^{\pm.001}$ | $\underline{3.113^{\pm.001}}$ | $\underline{9.410^{\pm.049}}$ | $1.553^{\pm.042}$ |
| T2M-GPT [61] | $0.492^{\pm.003}$ | $0.679^{\pm.002}$ | $0.775^{\pm.002}$ | $0.141^{\pm.005}$ | $3.121^{\pm.009}$ | $9.722^{\pm.082}$ | $1.831^{\pm.048}$ |
| GraphMotion (step=50) | $\underline{0.496^{\pm.003}}$ | $\underline{0.686^{\pm.003}}$ | $0.778^{\pm.002}$ | $\underline{0.118^{\pm.008}}$ | $3.143^{\pm.009}$ | $9.796^{\pm.069}$ | $2.603^{\pm.095}$ |
| GraphMotion (step=150) | $\mathbf{0.504^{\pm.003}}$ | $\mathbf{0.699^{\pm.002}}$ | $\mathbf{0.785^{\pm.002}}$ | $\mathbf{0.116^{\pm.007}}$ | $\mathbf{3.070^{\pm.008}}$ | $9.692^{\pm.067}$ | $\underline{2.766^{\pm.096}}$ |

Table 2: **Comparisons to current state-of-the-art methods on the KIT test set.** We repeat all the evaluations 20 times and report the average with a 95% confidence interval.

| Methods | R-Precision ↑ | | | FID ↓ | MM-Dist ↓ | Diversity → | MModality ↑ |
|---|---|---|---|---|---|---|---|
| | Top-1 | Top-2 | Top-3 | | | | |
| Real motion | $0.424^{\pm.005}$ | $0.649^{\pm.006}$ | $0.779^{\pm.006}$ | $0.031^{\pm.004}$ | $2.788^{\pm.012}$ | $11.08^{\pm.097}$ | - |
| Seq2Seq [44] | $0.103^{\pm.003}$ | $0.178^{\pm.005}$ | $0.241^{\pm.006}$ | $24.86^{\pm.348}$ | $7.960^{\pm.031}$ | $6.744^{\pm.106}$ | - |
| Text2Gesture [5] | $0.156^{\pm.004}$ | $0.255^{\pm.004}$ | $0.338^{\pm.005}$ | $12.12^{\pm.183}$ | $6.946^{\pm.029}$ | $9.334^{\pm.079}$ | - |
| MDM [54] | $0.164^{\pm.004}$ | $0.291^{\pm.004}$ | $0.396^{\pm.004}$ | $0.497^{\pm.021}$ | $9.191^{\pm.022}$ | $10.85^{\pm.109}$ | $1.907^{\pm.214}$ |
| Language2Pose [2] | $0.221^{\pm.005}$ | $0.373^{\pm.004}$ | $0.483^{\pm.005}$ | $6.545^{\pm.072}$ | $5.147^{\pm.030}$ | $9.073^{\pm.100}$ | - |
| Hier [12] | $0.255^{\pm.006}$ | $0.432^{\pm.007}$ | $0.531^{\pm.007}$ | $5.203^{\pm.107}$ | $4.986^{\pm.027}$ | $9.563^{\pm.072}$ | $2.090^{\pm.083}$ |
| TM2T [15] | $0.280^{\pm.005}$ | $0.463^{\pm.006}$ | $0.587^{\pm.005}$ | $3.599^{\pm.153}$ | $4.591^{\pm.026}$ | $9.473^{\pm.117}$ | $3.292^{\pm.081}$ |
| TEMOS [42] | $0.353^{\pm.006}$ | $0.561^{\pm.007}$ | $0.687^{\pm.005}$ | $3.717^{\pm.051}$ | $3.417^{\pm.019}$ | $10.84^{\pm.100}$ | $0.532^{\pm.034}$ |
| T2M [14] | $0.370^{\pm.005}$ | $0.569^{\pm.007}$ | $0.693^{\pm.007}$ | $2.770^{\pm.109}$ | $3.401^{\pm.008}$ | $10.91^{\pm.119}$ | $1.482^{\pm.065}$ |
| MLD [8] | $0.390^{\pm.008}$ | $0.609^{\pm.008}$ | $0.734^{\pm.007}$ | $0.404^{\pm.027}$ | $3.204^{\pm.027}$ | $10.80^{\pm.117}$ | $2.192^{\pm.071}$ |
| T2M-GPT [61] | $0.416^{\pm.006}$ | $0.627^{\pm.006}$ | $0.745^{\pm.006}$ | $0.514^{\pm.029}$ | $\underline{3.007^{\pm.023}}$ | $10.92^{\pm.108}$ | $1.570^{\pm.039}$ |
| MotionDiffuse [62] | $\underline{0.417^{\pm.004}}$ | $0.621^{\pm.004}$ | $0.739^{\pm.004}$ | $1.954^{\pm.062}$ | $\mathbf{2.958^{\pm.005}}$ | $\mathbf{11.10^{\pm.143}}$ | $0.730^{\pm.013}$ |
| GraphMotion (step=50) | $\underline{0.417^{\pm.008}}$ | $\underline{0.635^{\pm.006}}$ | $\underline{0.755^{\pm.004}}$ | $\mathbf{0.262^{\pm.021}}$ | $3.085^{\pm.031}$ | $11.21^{\pm.106}$ | $3.568^{\pm.132}$ |
| GraphMotion (step=150) | $\mathbf{0.429^{\pm.007}}$ | $\mathbf{0.648^{\pm.006}}$ | $\mathbf{0.769^{\pm.006}}$ | $\underline{0.313^{\pm.013}}$ | $3.076^{\pm.022}$ | $\underline{11.12^{\pm.135}}$ | $\mathbf{3.627^{\pm.113}}$ |

Finally, the training objective of the diffusion models can be defined as:

$$\mathcal{L}_{\text{DIF}} = \mathbb{E}_{\epsilon \sim \mathcal{N}(0,1), z \sim q(z|\mathcal{V}), t} \Big[ \underbrace{\|\epsilon^m - \phi_m(z^m, t^m, \mathcal{V}^m)\|_2^2}_{\text{Motion level}} + \underbrace{\|\epsilon^a - \phi_a(z^a, t^a, [\mathcal{V}^m, \mathcal{V}^a, z^m])\|_2^2}_{\text{Action level}}$$
$$+ \underbrace{\|\epsilon^s - \phi_s(z^s, t^s, [\mathcal{V}^m, \mathcal{V}^a, \mathcal{V}^s, z^a])\|_2^2}_{\text{Specific level}} \Big]. \tag{6}$$

During the training stage, the motion variational autoencoders are frozen. During the inference stage, the fine-grained semantic layer generates results based on the results from the coarse-grained semantic layer. We take the output at the specific level as the final result and use the motion decoder to decode the latent representation into the motion sequence.

It is worth noting that our method is as efficient as the one-stage diffusion methods during the inference stage, even though we decompose the diffusion process into three parts. This is because we can control the total number $T^m + T^a + T^s$ of iterations by restricting it to be the same as those of the one-stage diffusion methods, which we will discuss in experiments (see Tab. 5).

## 4 Experiments

**Datasets, Metrics and Implementation Details.** *Datasets.* We compare the proposed method with other methods on two commonly used public benchmarks: HumanML3D [14] and KIT [43].

| Real | **Ours** | MDM (ICLR23) | MLD (CVPR23) | MotionDiffuse (arXiv22) |
|:----:|:----:|:----:|:----:|:----:|

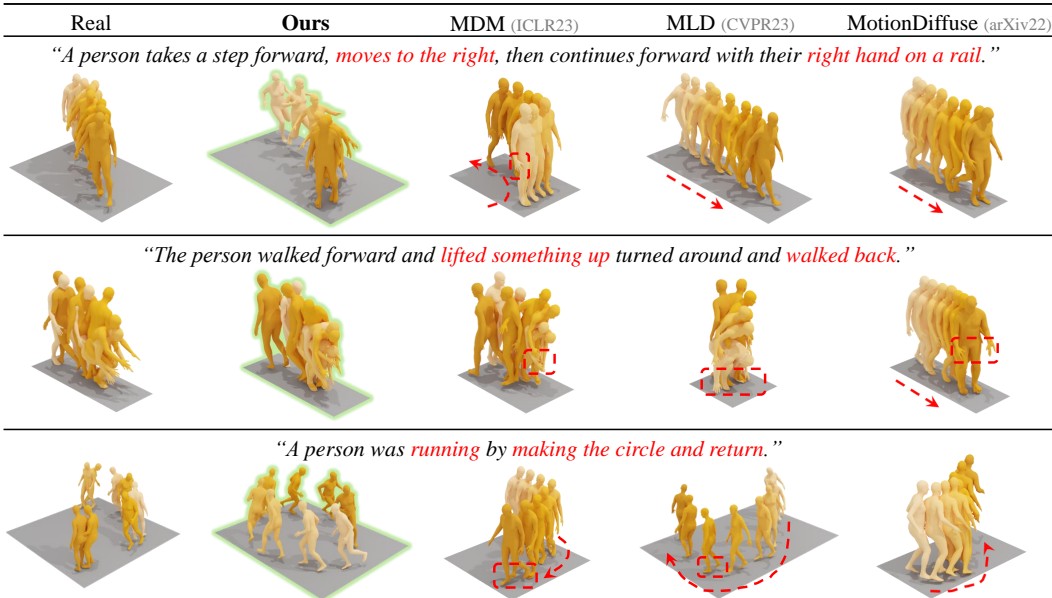

*"A person takes a step forward, moves to the right, then continues forward with their right hand on a rail."*

*"The person walked forward and lifted something up turned around and walked back."*

*"A person was running by making the circle and return."*

Figure 3: **Qualitative comparison of the existing methods.** We provide the motion results from three text prompts. The darker colors indicate the later in time. The generated results of our method better match the descriptions, while others have downgraded motions or improper semantics, demonstrating that our method achieves superior controllability compared to well-designed baseline models.

**HumanML3D** [14] is currently the largest 3D human motion dataset that originates from and textually reannotates the HumanAct12 [16] and AMASS [40] datasets. This dataset comprises 14,616 human motions and 44,970 text descriptions, with each motion accompanied by at least three precise descriptions. The lengths of these descriptions are around 12 words. **KIT** [43] contains 3,911 human motion sequences and 6,278 textual annotations. Each motion sequence is accompanied by one to four sentences, with an average description length of 8 words.

*Metrics.* Following previous works, we use the following five metrics to measure the performance of the model. (1) **R-Precision.** Under the feature space of the pre-trained network in [14], given one motion sequence and 32 text descriptions (1 ground-truth and 31 randomly selected mismatched descriptions), motion-retrieval precision calculates the text and motion Top 1/2/3 matching accuracy. (2) **Frechet Inception Distance (FID).** We measure the distribution distance between the generated and real motion using FID [17] on the extracted motion features [14]. (3) **Multimodal Distance (MM-Dist).** We calculate the average Euclidean distances between each text feature and the generated motion feature from that text. (4) **Diversity.** All generated motions are randomly sampled to two subsets of the same size. Then, we extract motion features [14] and compute the average Euclidean distances between the two subsets. (5) **Multimodality (MModality).** For each text description, we generate 20 motion sequences, forming 10 pairs of motions. We extract motion features and calculate the average Euclidean distance between each pair. We report the average of all text descriptions.

*Implementation Details.* For the motion variational autoencoder, motion encoder $\mathcal{E}$ and decoder $\mathcal{D}$ all consist of 9 layers and 4 heads with skip connection [46]. Following MLD [8], we utilize a frozen text encoder of the CLIP-ViT-L-14 [45] model for text representation. The dimension of node representation $D$ is set to 768. The dimension of latent embedding $D'$ is set to 256. We set the token sizes $C^m$ to 2, $C^a$ to 4, and $C^s$ to 8. We set $\lambda$ to 1e-4. All our models are trained with the AdamW [26, 38] optimizer using a fixed learning rate of 1e-4. We use 4 Tesla V100 GPUs for the training, and there are 128 samples on each GPU, so the total batch size is 512. For the HumanML3D dataset, the model is trained for 6,000 epochs during the motion variational autoencoder stage and 3,000 epochs during the diffusion stage. The number of diffusion steps of each level is 1,000 during training, and the step sizes $\beta_t$ are scaled linearly from 8.5×1e-4 to 0.012. For runtime, training tasks 16 hours for motion variational autoencoder and 24 hours for denoiser on 4 Tesla V100 GPUs.

**Comparisons to State-of-the-art.** We compare the proposed GraphMotion with other methods on two benchmarks. In Tab. 1, we show the results on the HumanML3D test set. Tab. 2 shows the results

Table 3: **Ablation study about each part of our method on the HumanML3D test set.** "↑" denotes that higher is better. "↓" denotes that lower is better.

| Semantic Graph | Graph Reasoning | Coarse-to-Fine Diffusion | R-Precision Top-1 ↑ | FID ↓ |
|:---:|:---:|:---:|:---:|:---:|
| | | | $0.485^{\pm.003}$ | $0.418^{\pm.013}$ |
| ✓ | | | $0.491^{\pm.003}$ | $0.281^{\pm.011}$ |
| ✓ | ✓ | | $0.494^{\pm.003}$ | $0.212^{\pm.015}$ |
| ✓ | | ✓ | $0.490^{\pm.004}$ | $0.196^{\pm.012}$ |
| ✓ | ✓ | ✓ | $\mathbf{0.504^{\pm.003}}$ | $\mathbf{0.116^{\pm.007}}$ |

Table 4: **Ablation study of the coarse-to-fine motion diffusion model on the HumanML3D test set.** "↑" denotes that higher is better. "↓" denotes that lower is better.

| Motion level | Action level | Specific level | R-Precision Top-1 ↑ | FID ↓ |
|:---:|:---:|:---:|:---:|:---:|
| | | | $0.485^{\pm.003}$ | $0.418^{\pm.013}$ |
| ✓ | | | $0.488^{\pm.003}$ | $0.217^{\pm.009}$ |
| ✓ | ✓ | | $0.494^{\pm.003}$ | $0.150^{\pm.011}$ |
| ✓ | ✓ | ✓ | $\mathbf{0.504^{\pm.003}}$ | $\mathbf{0.116^{\pm.007}}$ |

Table 5: **Ablation study about the total number of diffusion steps on the HumanML3D test set.** "↑" denotes that higher is better. "↓" denotes that lower is better. We repeat all the evaluations 20 times and report the average with a 95% confidence interval. "✘" denotes that this method does not apply this parameter. To speed up the sampling process, we use DDIM in practice following MLD.

| Methods | Diffusion Steps | | | R-Precision ↑ | | | FID ↓ |
|:---|:---:|:---:|:---:|:---:|:---:|:---:|:---:|
| | Motion $T^m$ | Action $T^a$ | Specific $T^s$ | Top-1 | Top-2 | Top-3 | |
| *The total number of diffusion steps is 1000 with DDPM [18]* | | | | | | | |
| MDM [54] | 1000 | ✘ | ✘ | $0.320^{\pm.005}$ | $0.498^{\pm.004}$ | $0.611^{\pm.007}$ | $0.544^{\pm.044}$ |
| MotionDiffuse [62] | 1000 | ✘ | ✘ | $0.491^{\pm.001}$ | $0.681^{\pm.001}$ | $0.782^{\pm.001}$ | $0.630^{\pm.001}$ |
| *The total number of diffusion steps is 50 with DDIM [50]* | | | | | | | |
| MLD [8] | 50 | ✘ | ✘ | $0.481^{\pm.003}$ | $0.673^{\pm.003}$ | $0.772^{\pm.002}$ | $0.473^{\pm.013}$ |
| GraphMotion (Ours) | 20 | 15 | 15 | $0.489^{\pm.003}$ | $0.676^{\pm.002}$ | $0.771^{\pm.002}$ | $0.131^{\pm.007}$ |
| GraphMotion (Ours) | 15 | 15 | 20 | $\mathbf{0.496^{\pm.003}}$ | $\mathbf{0.686^{\pm.003}}$ | $\mathbf{0.778^{\pm.002}}$ | $\mathbf{0.118^{\pm.008}}$ |
| *The total number of diffusion steps is 150 with DDIM [50]* | | | | | | | |
| MLD [8] | 150 | ✘ | ✘ | $0.461^{\pm.002}$ | $0.649^{\pm.003}$ | $0.797^{\pm.002}$ | $0.457^{\pm.011}$ |
| GraphMotion (Ours) | 50 | 50 | 50 | $\mathbf{0.504^{\pm.003}}$ | $\mathbf{0.699^{\pm.002}}$ | $0.785^{\pm.002}$ | $\mathbf{0.116^{\pm.007}}$ |

on the KIT test set. Our model consistently outperforms state-of-the-art methods on both benchmarks, which validates the effectiveness of our method. Moreover, we provide qualitative motion results in Fig. 3. Compared to other methods, our method generates motions that match the text descriptions better, indicating that our method is more sensitive to subtle differences in texts.

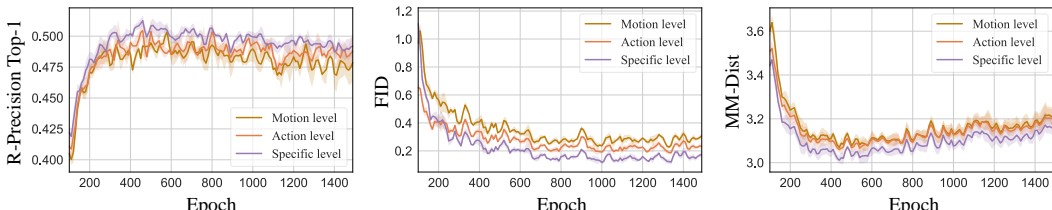

Figure 4: **Performance at each level on the HumanML3D test set.** We repeat all experiments 10 times and report the average with a 95% confidence interval. The performance of the specific level is the best, which confirms the effectiveness of the coarse-to-fine motion diffusion model.

**Ablative Analysis.** *Effect of each part of our method.* To explore the impact of each part of our method, we provide the ablation results in Tab. 3. "Semantic Graph" refers to the decomposition of the original motion description into nodes of three levels, namely motion, action, and specific. These nodes are then directly fed into the one-stage motion diffusion model to generate the final motion. "Graph Reasoning" utilizes a graph network to reason and update nodes in the hierarchical semantic graph. "Coarse-to-Fine Diffusion" decomposes the text-to-motion diffusion process into three semantic levels, from coarse to fine, which capture the overall motion, local actions, and action specifics, respectively. If not otherwise specified, all ablation experiments are performed using a diffusion step setting of 150. As shown in Tab. 3, we find that "Semantic Graph" can avoid the imbalanced learning of different textual components, thus significantly improving the semantic understanding ability of the model and the overall quality of motion generation. "Graph Reasoning" further enhances the semantic reasoning ability of the model. In addition, "Coarse-to-Fine Diffusion" can significantly improve the quality of motion generation. Our full model achieves the best performance, which demonstrates that the three parts are beneficial for motion generation.

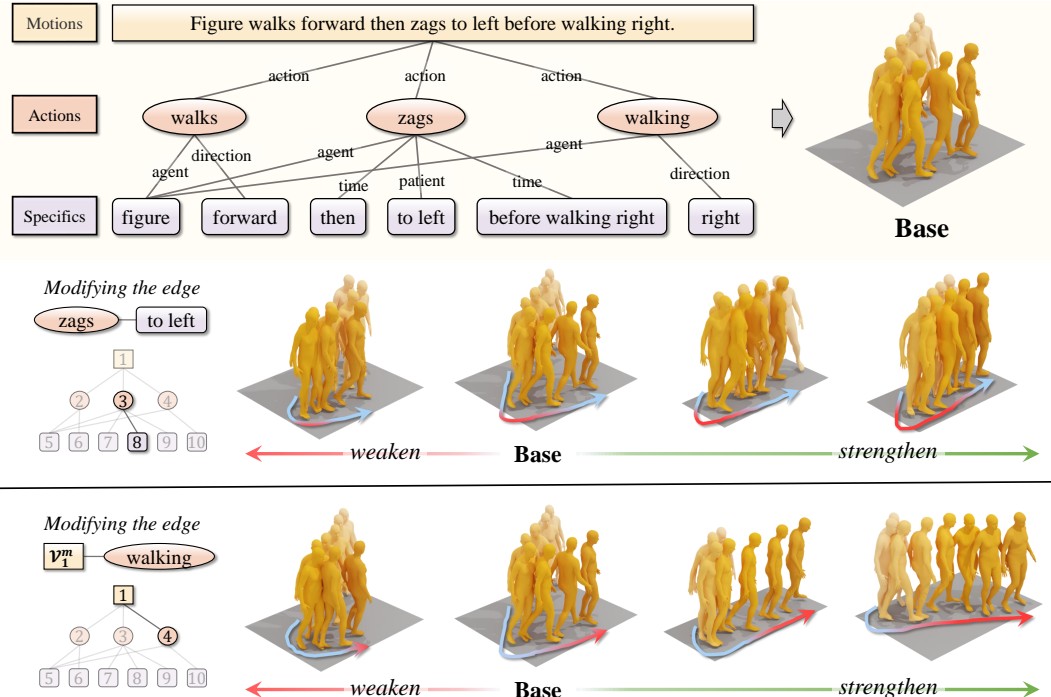

Figure 5: **Qualitative analysis of refining motion results.** The darker colors indicate the later in time. The trajectory of the human body is indicated by an arrow. The trajectory associated with the modified edge is highlighted in red, and other parts are identified in blue.

Table 6: **Quantitative experiment of the imbalance problem on the HumanML3D test set.** "↑" denotes that higher is better. "↓" denotes that lower is better.

| Methods | FID ↓ | MM-Dist ↓ | Diversity → | MModality ↑ |
|---|---|---|---|---|
| MDM | 5.622 | 7.163 | 8.713 | 3.578 |
| MLD | 3.492 | 5.632 | 8.874 | 3.596 |
| GraphMotion | **1.826** | **5.530** | **9.284** | **3.699** |

Table 7: **User studies for quantitative comparison.** We show the preference rate of GraphMotion over the compared model.

| Methods Compared | Preference Rate |
|---|---|
| GraphMotion vs. MotionDiffuse | 64.10% |
| GraphMotion vs. MLD | 56.41% |
| GraphMotion vs. Ground Truth | 48.72% |

*Analysis of the coarse-to-fine motion diffusion model.* In Tab. 4, we provide the ablation study of the coarse-to-fine motion diffusion model on the HumanML3D test set. These results prove that coarse-to-fine generation is beneficial to motion generation. In addition, we show the performance at each level in Fig. 4. Among the three levels, the performance of the specific level is the best, which confirms the effectiveness of the coarse-to-fine motion diffusion model.

*Effect of the diffusion steps.* In Tab. 5, we show the ablation study of the total number of diffusion steps on the HumanML3D test set. Following MLD [8], we adopt the denoising diffusion implicit models [50] (DDIM) during interference. As shown in Tab. 5, our method consistently outperforms the existing state-of-the-art methods with the same total number of diffusion steps, which demonstrates the efficiency of our method. With the increase of the total diffusion steps, the performance of our method is further improved, while the performance of MLD saturates. We find that the number of diffusion steps at the higher level (e.g., specific level) has a greater impact on the result. Therefore, in scenarios requiring high efficiency, we recommend allocating more diffusion steps to the higher level.

**Quantitative and Qualitative Discussion.** *Quantitative experiment of the imbalance problem.* In this experiment, we mask the verbs and action names in the motion description to force the model to generate motion only from action specifics. For example, given the motion description "a person walks several steps forward in a straight line.", we would mask "walks". Transformer extracts text features automatically and implicitly. However, it may encourage the model to take shortcuts, such as overemphasizing the action name "walks" at the expense of other important properties. Therefore, when the verbs and action names are masked, the other models, which directly use the transformer

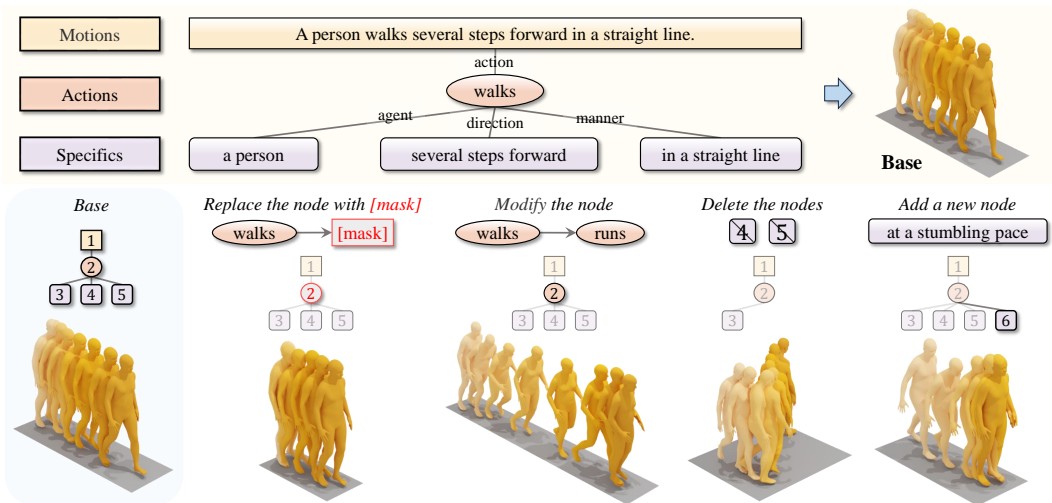

Figure 6: **Additional qualitative analysis of refining motion results.** The qualitative results demonstrate that our approach provides a novel method of refining generated motions.

to extract text features, fail to generate motion well. By contrast, the hierarchical semantic graph explicitly extracts the action specifics. The explicit factorization of the language embedding space facilitates a comprehensive understanding of motion description. It allows the model to infer from action specifics such as "several steps forward" and "in a straight line" that the overall motion is "walking forward". As shown in Tab. 6, our method can synthesize motion by relying only on action specifics, while other methods fail to generate motion well. These results indicate that our method avoids the imbalance problem of other methods.

*Human evaluation.* In our evaluation, we randomly selected 39 motion descriptions for the user study. As shown in Tab. 7, GraphMotion is preferred over the other models most of the time.

*Qualitative analysis of refining motion results.* To fine-tune the generated results for more fine-grained control, our method can continuously refine the generated motion by modifying the edge weights of the hierarchical semantic graph. As illustrated in Fig. 5, we can alter the action attributes by manipulating the weights of the edges of the action node and the specific node. For example, by increasing the weights of the edges of "zags" and "to left", the human body will move farther to the left. Moreover, by fine-tuning the weights of the edges of the global motion node and the action node, we can fine-tune the duration of the corresponding action in the whole motion. For example, by enhancing the weights of the edges of the global motion node and "walking", the length of the walk will be increased. In Fig. 6, we provide additional qualitative analysis of refining motion results. Specifically, we perform the additional operations on the hierarchical semantic graphs: (1) masking the node by replacing it with the MASK token; (2) modifying the node; (3) deleting nodes; (4) adding a new node. The qualitative results demonstrate that our approach provides a novel method of refining generated motions, which may have a far-reaching impact on the community.

## 5 Conclusion

In this paper, we focus on improving the controllability of text-driven human motion generation. To provide fine-grained control over motion details, we propose a novel control signal called the hierarchical semantic graph, which consists of three kinds of abstract nodes, namely motions, actions, and specifics. Correspondingly, to leverage the coarse-to-fine topology of hierarchical semantic graphs, we decompose the text-to-motion diffusion process into three semantic levels, which correspond to capturing the overall motion, local actions, and action specifics. Extensive experiments demonstrate that our method achieves better controllability than the existing state-of-the-art methods. More encouragingly, our method can continuously refine the generated motion by modifying the edge weights of hierarchical semantic graphs, which may have a far-reaching impact on the community.

**Acknowledgements.** This work was supported by the National Key R&D Program of China (2022ZD0118101), Nature Science Foundation of China (No.62202014), and Shenzhen Basic Research Program (No.JCYJ20220813151736001).

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

**Abstract**   This appendix provides additional discussions (Sec. A), implementation details (Sec. B), more qualitative results (Sec. C), several additional experiments (Sec. D), details of motion representations and metric definitions (Sec. E).

**Code**   Code is available at https://github.com/jpthu17/GraphMotion. In this code, we provide the process of the training and evaluation of the proposed method, and the pre-trained weights.

# A   Additional Discussions

## A.1   Potential Negative Societal Impacts

While our work effectively enhances the quality of human motion synthesis, there is a potential risk that it may be used for generating fake content, such as generating fake news, which can pose a threat to information security. Moreover, when factoring in energy consumption, there is a possibility that the widespread use of generative models for synthesizing human motions may contribute to increased carbon emissions and exacerbate global warming.

## A.2   Limitations of our Work

Although our method makes some progress, there are still many limitations worth further study. (1) The proposed GraphMotion inherits the randomness of diffusion models. This property benefits diversity but may yield undesirable results sometimes. (2) The human motion synthesis capabilities of GraphMotion are limited by the performance of the pre-trained motion variational autoencoders, which we will discuss in experiments (Tab. D and Tab. E). This defect also exists in the existing state-of-the-art methods, such as MLD [8] and T2M-GPT [61], which also use motion variational autoencoder. (3) Though the proposed GraphMotion brings negligible extra cost on computations, it is still limited by the slow inference speed of existing diffusion models. We will discuss the inference time in experiments (Tab. C). This defect also exists in the existing state-of-the-art methods, such as MDM [54] and MLD [8], which also use diffusion models.

## A.3   Future Work

In this paper, we focus on improving the controllability of text-driven human motion generation. Recently, large language models have made remarkable progress, making large language models a promising text extractor for human motion generation. However, despite their strengths in general reasoning and broad applicability, large language models may not be optimized for extracting subtle motion nuances. In future research, we will incorporate the features of large-scale languages into our model, using hierarchical semantic graphs to give large language models the ability to extract fine-grained motion description structures.

# B   Implementation Details

## B.1   Details of Hierarchical Semantic Graphs

To obtain actions, attributes of action as well as the semantic role of each attribute to the corresponding action, we implement a semantic parser of motion descriptions based on a semantic role parsing toolkit [48, 7]. Specifically, given the motion description, the parser extracts verbs that appeared in the sentence and attribute phrases corresponding verb, and the semantic role of each attribute phrase. The overall sentence is treated as the global motion node in the hierarchical graph. The verbs are considered as action nodes and connected to the motion node with direct edges, allowing for implicit learning of the temporal relationships among various actions during graph reasoning. The attribute phrases are specific nodes that are connected with action nodes. The edge type between action and specific nodes is determined by the semantic role of the specifics in relation to the action.

Table A: **Node types and edge types in the parsed hierarchical semantic graph.** Each edge type corresponds to a type of semantic role.

| Node type | Description |
|---|---|
| Motion | global motion description |
| Action | verb |
| Specific | attribute of action |

| Edge type | Description |
|---|---|
| ARG0 | agent |
| ARG1 | patient |
| ARG2 | instrument, benefactive |
| ARG3 | start point |
| ARG4 | end point |
| ARGM-LOC | location (where) |
| ARGM-MNR | manner (how) |
| ARGM-TMP | time (when) |
| ARGM-DIR | direction (where to/from) |
| ARGM-ADV | miscellaneous |
| ARGM-MA | motion-action dependencies |
| OTHERS | other argument types, e.g., action |

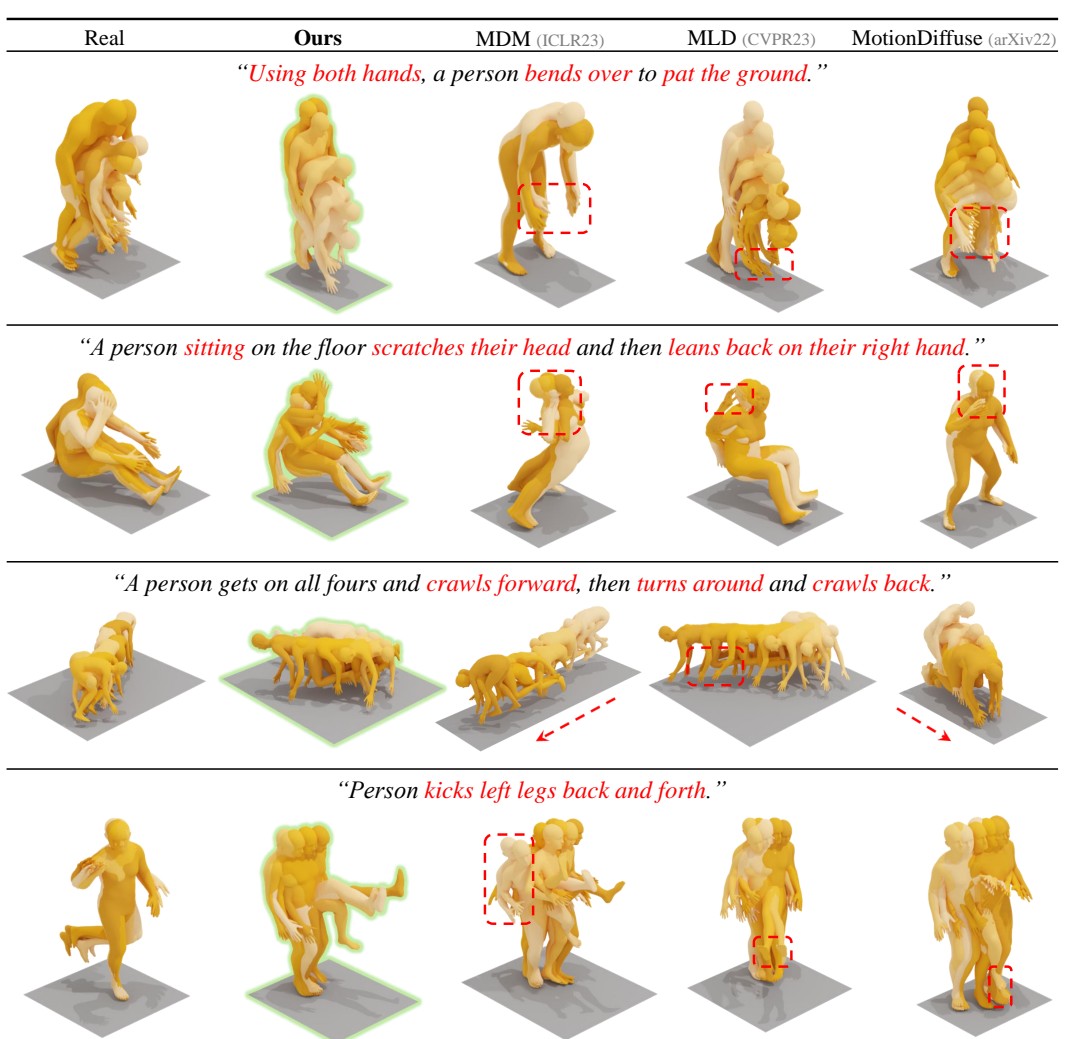

| Real | **Ours** | MDM (ICLR23) | MLD (CVPR23) | MotionDiffuse (arXiv22) |

*"Using both hands, a person bends over to pat the ground."*

*"A person sitting on the floor scratches their head and then leans back on their right hand."*

*"A person gets on all fours and crawls forward, then turns around and crawls back."*

*"Person kicks left legs back and forth."*

Figure A: **Qualitative comparison of the existing methods.** The darker colors indicate the later in time. The generated results of our method better match the descriptions, while others have downgraded motions or improper semantics, demonstrating that our method achieves superior controllability compared to well-designed baseline models. We have provided a supplemental video in our supplementary material. In the supplemental video, we show comparisons of text-driven motion generation. We suggest the reader watch this video for dynamic motion results.

As shown in Tab. A, we extract three types (motions, actions, and specifics) of nodes and twelve types of edges to represent various associations among the nodes.

## B.2 Classifier-free Diffusion Guidance

Following MLD [8], our denoiser network is learned with classifier-free diffusion guidance [19]. The classifier-free diffusion guidance improves the quality of samples by reducing diversity in conditional diffusion models. Concretely, it learns both the conditioned and the unconditioned distribution (10% dropout [51]) of the samples. Finally, we perform a linear combination in the following manner, which is formulated as:

$$
\begin{aligned}
\widehat{\epsilon^m}_{scale} &= \alpha^{'}\phi_m(z^m, t^m, \mathcal{V}^m) + (1-\alpha^{'})\phi_m(z^m, t^m, \varnothing), \\
\widehat{\epsilon^a}_{scale} &= \alpha^{'}\phi_a(z^a, t^a, [\mathcal{V}^m, \mathcal{V}^a, z^m]) + (1-\alpha^{'})\phi_a(z^a, t^a, \varnothing), \\
\widehat{\epsilon^s}_{scale} &= \alpha^{'}\phi_s(z^s, t^s, [\mathcal{V}^m, \mathcal{V}^a, \mathcal{V}^s, z^a]) + (1-\alpha^{'})\phi_s(z^s, t^s, \varnothing),
\end{aligned}
\tag{A}
$$

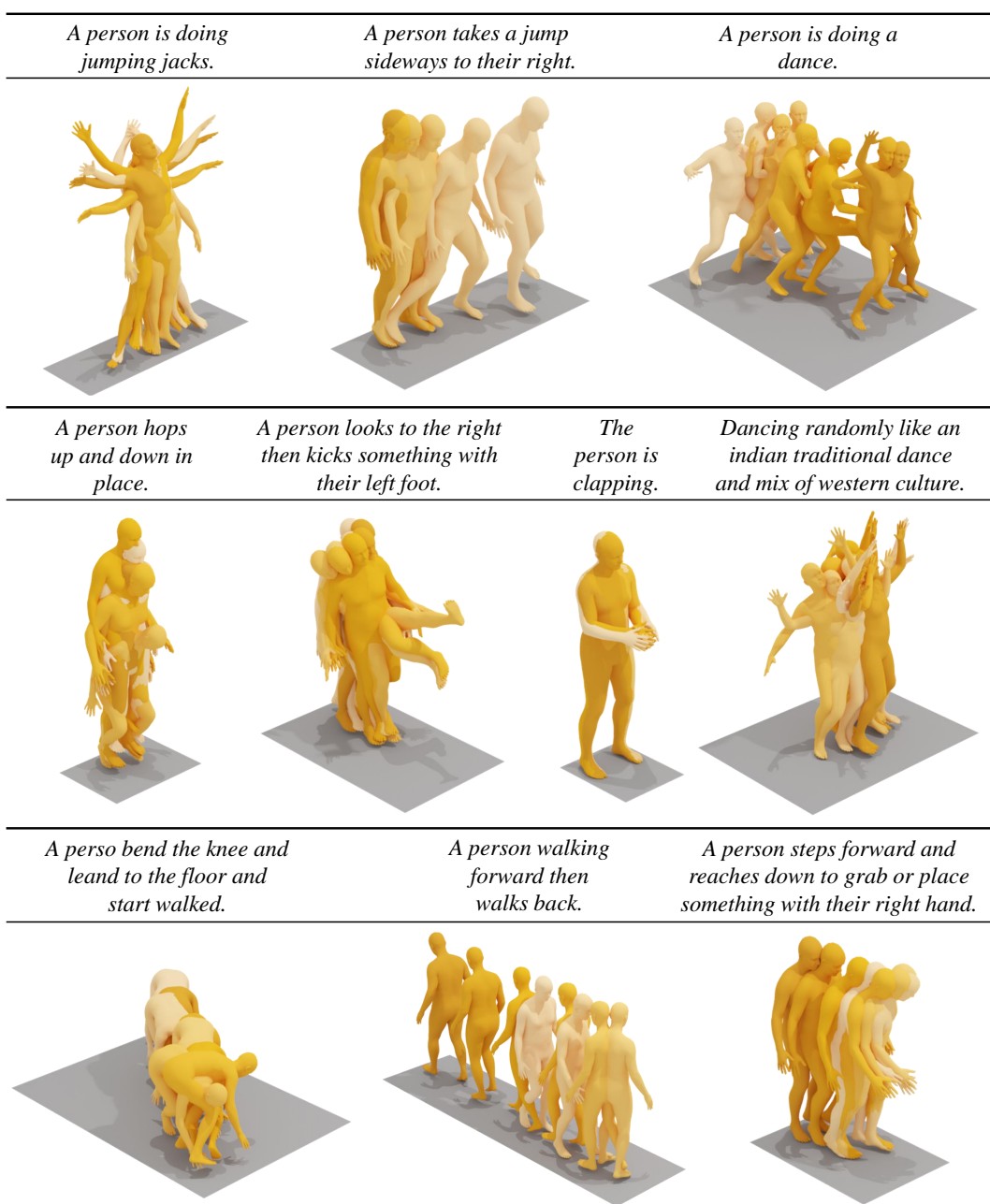

Figure B: **Additional qualitative motion results are generated with text prompts of the HumanML3D test set.** The darker colors indicate the later in time. These results demonstrate that our method can generate diverse and accurate motion sequences.

Where $\alpha'$ is the guidance scale and $\alpha' > 1$ can strengthen the effect of guidance [8]. We set $\alpha'$ to 7.5 in practice following MLD. Please refer to our code for more details.

### B.3 Implementation Details for Different Datasets

Following MLD [8], we utilize a frozen text encoder of the CLIP-ViT-L-14 [45] model for text representation. The dimension of node representation $D$ is set to 768. The dimension of latent embedding $D'$ is set to 256. For the motion variational autoencoder, motion encoder $\mathcal{E}$ and decoder $\mathcal{D}$ all consist of 9 layers and 4 heads with skip connection [46]. We set the token sizes $C^m$ to 2, $C^a$

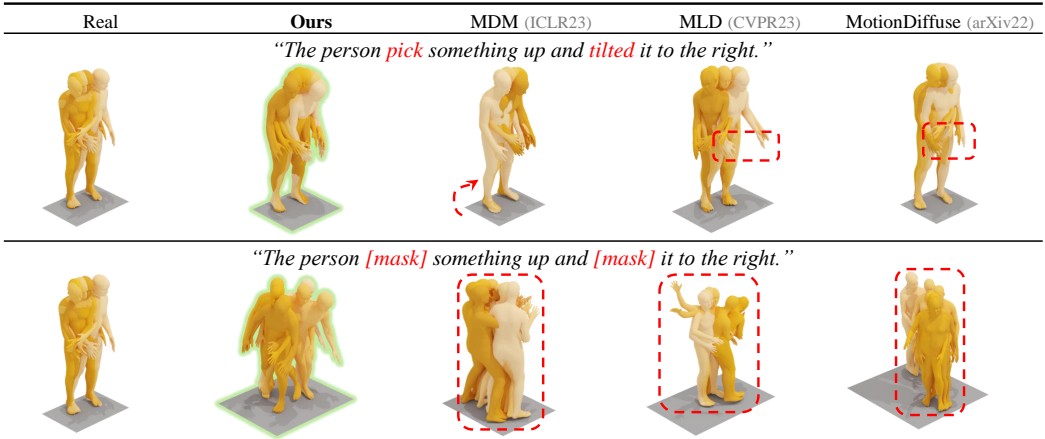

Figure C: **Qualitative analysis on the imbalance problem.** The darker colors indicate the later in time. When the verbs and action names are masked, existing models tend to generate motion randomly. In contrast, our method can generate motion based solely on the action specifics. These results show that our method is not overly focused on the verbs and action names.

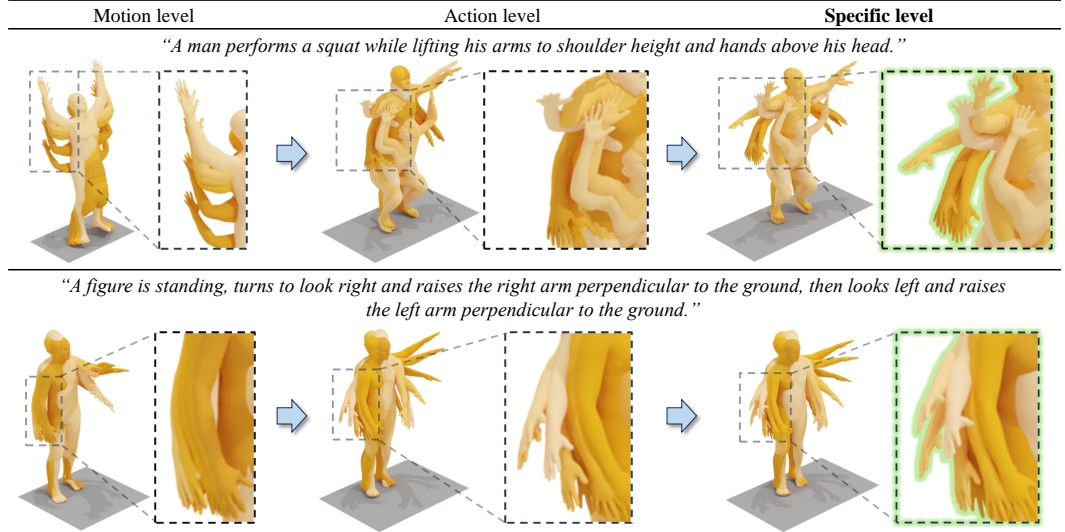

Figure D: **Qualitative comparison of different hierarchies.** The output at the higher level (e.g., specific level) has more action details. Specifically, the motion level generates only coarse-grained overall motion. The action level generates local actions better than the motion level but lacks action specifics. The specific level generates more action specifics than the action level.

to 4, and $C^s$ to 8. We set $\lambda$ to 1e-4. All our models are trained with the AdamW [26, 38] optimizer using a fixed learning rate of 1e-4. We use 4 Tesla V100 GPUs for the training, and there are 128 samples on each GPU, so the total batch size is 512. The number of diffusion steps of each level is 1,000 during training, and the step sizes $\beta_t$ are scaled linearly from 8.5×1e-4 to 0.012. We keep running a similar number of iterations on different data sets. For the HumanML3D dataset, the model is trained for 6,000 epochs during the motion variational autoencoder stage and 3,000 epochs during the diffusion stage. For the KIT dataset, the model is trained for 30,000 epochs during the motion variational autoencoder stage and 15,000 epochs during the diffusion stage. Code is available at https://github.com/jpthu17/GraphMotion. In this code, we provide the process of the training and evaluation of the proposed method, and the pre-trained model.

Table B: **Ablation study about the total number of diffusion steps on the HumanML3D test set.** "↑" denotes that higher is better. "↓" denotes that lower is better. We repeat all the evaluations 20 times and report the average with a 95% confidence interval. "✗" denotes that this method does not apply this parameter. To speed up the sampling process, we use DDIM in practice following MLD.

| Methods | Diffusion Steps | | | R-Precision ↑ | | | FID ↓ |
|---|---|---|---|---|---|---|---|
| | Motion $T^m$ | Action $T^a$ | Specific $T^s$ | Top-1 | Top-2 | Top-3 | |
| *The total number of diffusion steps is 1000 with DDPM [18]* | | | | | | | |
| MDM [54] | 1000 | ✗ | ✗ | $0.320^{\pm.005}$ | $0.498^{\pm.004}$ | $0.611^{\pm.007}$ | $0.544^{\pm.044}$ |
| MotionDiffuse [62] | 1000 | ✗ | ✗ | $0.491^{\pm.001}$ | $0.681^{\pm.001}$ | $0.782^{\pm.001}$ | $0.630^{\pm.001}$ |
| *The total number of diffusion steps is 50 with DDIM [50]* | | | | | | | |
| MLD [8] | 50 | ✗ | ✗ | $0.481^{\pm.003}$ | $0.673^{\pm.003}$ | $0.772^{\pm.002}$ | $0.473^{\pm.013}$ |
| GraphMotion (Ours) | 20 | 15 | 15 | $0.489^{\pm.003}$ | $0.676^{\pm.002}$ | $0.771^{\pm.002}$ | $0.131^{\pm.007}$ |
| GraphMotion (Ours) | 15 | 15 | 20 | $\mathbf{0.496^{\pm.003}}$ | $\mathbf{0.686^{\pm.003}}$ | $\mathbf{0.778^{\pm.002}}$ | $\mathbf{0.118^{\pm.008}}$ |
| *The total number of diffusion steps is 150 with DDIM [50]* | | | | | | | |
| MLD [8] | 150 | ✗ | ✗ | $0.461^{\pm.002}$ | $0.649^{\pm.003}$ | $0.797^{\pm.002}$ | $0.457^{\pm.011}$ |
| GraphMotion (Ours) | 50 | 50 | 50 | $\mathbf{0.504^{\pm.003}}$ | $\mathbf{0.699^{\pm.002}}$ | $\mathbf{0.785^{\pm.002}}$ | $\mathbf{0.116^{\pm.007}}$ |
| *The total number of diffusion steps is 300 with DDIM [50]* | | | | | | | |
| MLD [8] | 300 | ✗ | ✗ | $0.473^{\pm.002}$ | $0.664^{\pm.003}$ | $0.765^{\pm.002}$ | $0.403^{\pm.011}$ |
| GraphMotion (Ours) | 100 | 100 | 100 | $\mathbf{0.486^{\pm.003}}$ | $\mathbf{0.671^{\pm.004}}$ | $\mathbf{0.767^{\pm.003}}$ | $\mathbf{0.096^{\pm.008}}$ |
| *The total number of diffusion steps is 1000 with DDIM [50]* | | | | | | | |
| MLD [8] | 1000 | ✗ | ✗ | $0.452^{\pm.002}$ | $0.639^{\pm.003}$ | $0.751^{\pm.002}$ | $0.460^{\pm.013}$ |
| GraphMotion (Ours) | 400 | 300 | 300 | $0.475^{\pm.003}$ | $0.659^{\pm.003}$ | $0.756^{\pm.003}$ | $0.136^{\pm.007}$ |
| GraphMotion (Ours) | 300 | 300 | 400 | $\mathbf{0.484^{\pm.003}}$ | $\mathbf{0.694^{\pm.003}}$ | $\mathbf{0.787^{\pm.003}}$ | $\mathbf{0.132^{\pm.008}}$ |

# C    Additional Qualitative Analysis

## C.1    Qualitative Comparison of the Existing Methods

We provide additional qualitative motion results in Fig. A. Compared to other methods, our method generates motions that match the text descriptions better, indicating that our method is more sensitive to subtle differences in texts. The generated results demonstrate that our method achieves superior controllability compared to well-designed baseline models.

## C.2    Additional Visualization Results

In Fig. B, we provide additional qualitative motion results which are generated with text prompts of the HumanML3D test set. These results demonstrate that our method can generate diverse and accurate motion sequences from a variety of motion descriptions.

## C.3    Qualitative Analysis on the Imbalance Problem

To demonstrate the imbalance problem of other methods and prove that our method does not have this problem, we mask the verbs and action names in the motion description to force the model to generate motion only from action specifics. As shown in Fig. C, when the verbs and action names are masked, existing models tend to generate motion randomly. In contrast, our method can generate motion that matches the description based solely on the action specifics. These results show that our method is not overly focused on the verbs and action names.

## C.4    Qualitative Comparison of Different Hierarchies

We provide different levels of qualitative comparison in Fig. D. The results show that the output at the higher level (e.g., specific level) has more action details. Specifically, the motion level generates only coarse-grained overall motion. The action level generates local actions better than the motion level but lacks action specifics. The specific level generates more action specifics than the action level.

Table C: **Evaluation of Inference time costs on the HumanML3D test set.** We evaluate the average time per sample with different diffusion schedules and FID. "↓" denotes that lower is better. Please note the bad FID of MDM with DDIM is mentioned in their GitHub issues #76. "✗" denotes that this method does not apply this parameter. We use DDIM in practice following MLD.

| Methods | Reference | Diffusion Steps | | | Average time per sample (s) ↓ | FID ↓ |
| | | Motion $T^m$ | Action $T^a$ | Specific $T^s$ | | |
|---|---|---|---|---|---|---|
| *The total number of diffusion steps is 1000 with DDPM [18]* | | | | | | |
| MDM [54] | ICLR 2023 | 1000 | ✗ | ✗ | 178.7699 | **0.544** |
| MLD [8] | CVPR 2023 | 1000 | ✗ | ✗ | 5.5045 | 0.568 |
| *The total number of diffusion steps is 50 with DDIM [50]* | | | | | | |
| MDM [54] | ICLR 2023 | 50 | ✗ | ✗ | 20.5678 | 7.334 |
| MLD [8] | CVPR 2023 | 50 | ✗ | ✗ | 0.9349 | 0.473 |
| GraphMotion | Ours | 20 | 15 | 15 | 0.9094 | 0.131 |
| GraphMotion | Ours | 15 | 15 | 20 | 0.7758 | **0.118** |
| *The total number of diffusion steps is 150 with DDIM [50]* | | | | | | |
| MLD [8] | CVPR 2023 | 150 | ✗ | ✗ | 2.4998 | 0.457 |
| GraphMotion | Ours | 50 | 50 | 50 | 2.5518 | **0.116** |
| *The total number of diffusion steps is 1000 with DDIM [50]* | | | | | | |
| MLD [8] | CVPR 2023 | 1000 | ✗ | ✗ | 16.6654 | 0.460 |
| GraphMotion | Ours | 400 | 300 | 300 | 22.1238 | 0.136 |
| GraphMotion | Ours | 300 | 300 | 400 | 17.0912 | **0.132** |

# D  Additional Experiments

## D.1  Analysis of the Diffusion Steps

In Tab. B, we show the ablation study of the total number of diffusion steps on the HumanML3D test set. Following MLD [8], we adopt the denoising diffusion implicit models [50] (DDIM) during interference. As shown in Tab. B, our method consistently outperforms the existing state-of-the-art methods with the same total number of diffusion steps, which demonstrates the efficiency of our method. We find that the number of diffusion steps at the higher level (e.g., specific level) has a greater impact on the result. Therefore, in scenarios requiring high efficiency, we recommend allocating more diffusion steps to the higher level. Moreover, with the increase of the total diffusion steps, the performance of our method is further improved, while the performance of MLD saturates. These results further prove the superiority of our design.

## D.2  Analysis of the Inference Time

In Tab. C, we provide the evaluation of inference time costs. Our method is as efficient as the one-stage diffusion methods during the inference stage, even though we decompose the diffusion process into three parts. This is because we can control the total number $T^m + T^a + T^s$ of iterations by restricting it to be the same as those of the one-stage diffusion methods. As shown in Tab. C, the inference speed of our method is comparable to that of the existing state-of-the-art methods with the same total number of diffusion steps, which demonstrates the efficiency of our method.

## D.3  Analysis of the motion VAE models

We provide the evaluation of the motion VAE models. In Tab. D, we show the results on the HumanML3D test set. Tab. E shows the results on the KIT test set. Among the three levels, the performance of the specific level is the best, which indicates that increasing the token size can improve the reconstruction ability of the motion VAE models.

# E  Motion Representations and Metric Definitions

## E.1  Motion Representations

Motion representation can be summarized into the following four categories, and we follow the previous work of representing motion in latent space.

Table D: **Evaluation of the VAE models on the motion part of the HumanML3D test set.** "↑" denotes that higher is better. "↓" denotes that lower is better. "→" denotes that results are better if the metric is closer to the real motion. The performance of the specific level is the best.

| Methods | Token Size | R-Precision ↑ | | | FID ↓ | Diversity → |
| --- | --- | --- | --- | --- | --- | --- |
| | | Top-1 | Top-2 | Top-3 | | |
| Real motion | - | 0.511 | 0.703 | 0.797 | 0.002 | 9.503 |
| Motion Level | 2 | 0.498 | 0.692 | 0.791 | 1.906 | 9.675 |
| Action Level | 4 | 0.514 | 0.703 | 0.793 | 0.068 | **9.610** |
| Specific Level | 8 | **0.525** | **0.708** | **0.800** | **0.019** | 9.863 |

Table E: **Evaluation of the VAE models on the motion part of the KIT test set.** "↑" denotes that higher is better. "↓" denotes that lower is better. "→" denotes that results are better if the metric is closer to the real motion. The performance of the specific level is the best.

| Methods | Token Size | R-Precision ↑ | | | FID ↓ | Diversity → |
| --- | --- | --- | --- | --- | --- | --- |
| | | Top-1 | Top-2 | Top-3 | | |
| Real motion | - | 0.424 | 0.649 | 0.779 | 0.031 | 11.08 |
| Motion Level | 2 | **0.431** | 0.623 | 0.745 | 1.196 | 10.66 |
| Action Level | 4 | 0.413 | **0.644** | **0.770** | 0.396 | 10.85 |
| Specific Level | 8 | 0.414 | 0.640 | 0.760 | **0.361** | **10.86** |

**Latent Format.** Following previous works [42, 8, 61], we encode the motion into the latent space with a motion variational autoencoder [27]. The latent representation is formulated as:

$$\hat{x}^{1:L} = \mathcal{D}(z), \quad z = \mathcal{E}(x^{1:L}). \tag{B}$$

**HumanML3D Format.** HumanML3D [14] proposes a motion representation $x^{1:L}$ inspired by motion features in character control. This motion representation is well-suited for neural networks. To be specific, the $i_{th}$ pose $x^i$ is defined by a tuple consisting of the root angular velocity $r^a \in \mathbb{R}$ along the Y-axis, root linear velocities $(r^x, r^z \in \mathbb{R})$ on the XZ-plane, root height $r^y \in \mathbb{R}$, local joints positions $j^p \in \mathbb{R}^{3N_j}$, velocities $j^v \in \mathbb{R}^{3N_j}$, and rotations $j^r \in \mathbb{R}^{6N_j}$ in root space, and binary foot-ground contact features $c^f \in \mathbb{R}^4$ obtained by thresholding the heel and toe joint velocities. Here, $N_j$ denotes the joint number. Finally, the HumanML3D format can be defined as:

$$x^i = \{r^a, r^x, r^z, r^y, j^p, j^v, j^r, c^f\}. \tag{C}$$

**SMPL-based Format.** SMPL [37] is one of the most widely used parametric human models. SMPL and its variants propose motion parameters $\theta$ and shape parameters $\beta$. $\theta \in \mathbb{R}^{3 \times 23 + 3}$ is rotation vectors for 23 joints and a root, while $\beta$ represents the weights for linear blended shapes. The global translation $r$ is also incorporated to formulate the representation as follows:

$$x^i = \{r, \theta, \beta\}. \tag{D}$$

**MMM Format.** Master Motor Map [52] (MMM) representations propose joint angle parameters based on a uniform skeleton structure with 50 degrees of freedom (DoFs). In text-to-motion tasks, recent methods [2, 12, 42] converts joint rotation angles into $J = 21$ joint XYZ coordinates. Given the global trajectory $t_{root}$ and $p_m \in \mathbb{R}^{3J}$, the preprocessed representation is formulated as:

$$x^i = \{p_m, t_{root}\}. \tag{E}$$

## E.2 Metric Definitions

Following previous works, we use the following five metrics to measure the performance of the model. Note that global representations of motion and text descriptions are first extracted with the pre-trained network in [14].

**R-Precision.** Under the feature space of the pre-trained network in [14], given one motion sequence and 32 text descriptions (1 ground-truth and 31 randomly selected mismatched descriptions), motion-retrieval precision calculates the text and motion Top 1/2/3 matching accuracy.

**Frechet Inception Distance (FID).** We measure the distribution distance between the generated and real motion using FID [17] on the extracted motion features [14]. The FID is calculated as:

$$\text{FID} = \|\mu_{gt} - \mu_{pred}\|^2 - \text{Tr}(\Sigma_{gt} + \Sigma_{pred} - 2(\Sigma_{gt}\Sigma_{pred})^{\frac{1}{2}}), \tag{F}$$

where $\Sigma$ is the covariance matrix. Tr denotes the trace of a matrix. $\mu_{gt}$ and $\mu_{pred}$ are the mean of ground-truth motion features and generated motion features.

**Multimodal Distance (MM-Dist).** Given $N$ randomly generated samples, we calculate the average Euclidean distances between each text feature $f_t$ and the generated motion feature $f_m$ from that text. The multimodal distance is calculated as:

$$\text{MM-Dist} = \frac{1}{N} \sum_{i=1}^{N} \|f_{t,i} - f_{m,i}\|, \tag{G}$$

where $f_{t,i}$ and $f_{m,i}$ are the features of the $i_{th}$ text-motion pair.

**Diversity.** All generated motions are randomly sampled to two subsets ($\{x_1, x_2, ..., x_{X_d}\}$ and $\{x_1', x_2', ..., x_{X_d}'\}$) of the same size $X_d$. Then, we extract motion features [14] and compute the average Euclidean distances between the two subsets:

$$\text{Diversity} = \frac{1}{X_d} \sum_{i=1}^{X_d} \|x_i - x_i'\|. \tag{H}$$

**Multimodality (MModality).** We randomly sample a set of text descriptions with size $J_m$ from all descriptions. For each text description, we generate $2 \times X_m$ motion sequences, forming $X_m$ pairs of motions. We extract motion features and calculate the average Euclidean distance between each pair. We report the average of all text descriptions. We define features of the $j_{th}$ pair of the $i_{th}$ text description as $(x_{j,i}, x_{j,i}')$. The multimodality is calculated as:

$$\text{MModality} = \frac{1}{J_m \times X_m} \sum_{j=1}^{J_m} \sum_{i=1}^{X_m} \|x_{j,i} - x_{j,i}'\|. \tag{I}$$

