# Act As You Wish: Fine-Grained Control of Motion Diffusion Model with Hierarchical Semantic Graphs Supplementary Material

**Peng Jin**[1,4]    **Yang Wu**[3]*    **Yanbo Fan**[3]    **Zhongqian Sun**[3]    **Yang Wei**[3]    **Li Yuan**[1,2,4]*

[1] School of Electronic and Computer Engineering, Peking University, Shenzhen, China
[2] Peng Cheng Laboratory, Shenzhen, China    [3] Tencent AI Lab, China
[4]AI for Science (AI4S)-Preferred Program, Peking University Shenzhen Graduate School, China

jp21@stu.pku.edu.cn    dylan.yangwu@qq.com    yuanli-ece@pku.edu.cn

**Abstract**    This appendix provides additional discussions (Sec. A), implementation details (Sec. B), more qualitative results (Sec. C), several additional experiments (Sec. D), details of motion representations and metric definitions (Sec. E).

**Code**    Code is available at https://github.com/jpthu17/GraphMotion. In this code, we provide the process of the training and evaluation of the proposed method, and the pre-trained weights.

## A   Additional Discussions

### A.1   Potential Negative Societal Impacts

While our work effectively enhances the quality of human motion synthesis, there is a potential risk that it may be used for generating fake content, such as generating fake news, which can pose a threat to information security. Moreover, when factoring in energy consumption, there is a possibility that the widespread use of generative models for synthesizing human motions may contribute to increased carbon emissions and exacerbate global warming.

### A.2   Limitations of our Work

Although our method makes some progress, there are still many limitations worth further study. (1) The proposed GraphMotion inherits the randomness of diffusion models. This property benefits diversity but may yield undesirable results sometimes. (2) The human motion synthesis capabilities of GraphMotion are limited by the performance of the pre-trained motion variational autoencoders, which we will discuss in experiments (Tab. D and Tab. E). This defect also exists in the existing state-of-the-art methods, such as MLD [3] and T2M-GPT [26], which also use motion variational autoencoder. (3) Though the proposed GraphMotion brings negligible extra cost on computations, it is still limited by the slow inference speed of existing diffusion models. We will discuss the inference time in experiments (Tab. C). This defect also exists in the existing state-of-the-art methods, such as MDM [25] and MLD [3], which also use diffusion models.

### A.3   Future Work

In this paper, we focus on improving the controllability of text-driven human motion generation. Recently, large language models have made remarkable progress, making large language models a promising text extractor for human motion generation. However, despite their strengths in general reasoning and broad applicability, large language models may not be optimized for extracting subtle motion nuances. In future research, we will incorporate the features of large-scale languages into

---

*Corresponding author: Yang Wu, Li Yuan.

37th Conference on Neural Information Processing Systems (NeurIPS 2023).

our model, using hierarchical semantic graphs to give large language models the ability to extract fine-grained motion description structures. In addition, the application of hierarchical semantic graphs to other cross-modal tasks [10, 11, 15], such as cross-modal retrieval [9, 12] and visual question answering, is also a promising research direction.