# OpenReview forum: "Act As You Wish: Fine-Grained Control of Motion Diffusion Model with Hierarchical Semantic Graphs"
_NeurIPS.cc/2023/Conference — NeurIPS 2023 poster_

### Official Review · Reviewer_E76p · 2023-07-04

**Soundness:** 3 good
**Presentation:** 3 good
**Contribution:** 3 good
**Rating:** 6
**Confidence:** 4

**Summary:**

The authors identify two major issues of text-to-motion generation as overemphasis on action names and the coarseness of sentence-level representations.
To this end, hierarchical semantic graphs are adopted to factorize coarse sentences into fine-grained action concepts, thus refining the generated motion from coarse-to-fine.
Furthermore, the hierarchical design allows flexible control of the generation procedure.
Extensive experiments show the efficacy of the proposed method.

**Strengths:**

The proposed three tiers of node as motions, actions and specifics are interesting.

The hierarchical diffusion is well-designed in disentangling different level of granularity. Also, the experimental results show reasonable improvements with more and levels.

The performance is impressive.

The motion refinement application is attractive and promising.

The analysis on distribution of diffusion steps might be interesting in identifying the performance bottleneck among different semantic levels.

Extensive ablation experiments provide a nice view of the effectiveness of different components.

**Weaknesses:**

Qualitative comparison of different hierarchies is missed. It would be helpful to visualize motion generated from different level of latent embeddings, since fundamentally human is the only gold standard to evaluate motion generation, given the rather close numeric metrics of different levels.

The ablation on the design of the semantic graph is missed.

A more detailed description of the motion refining procedure, like a figure, an algorithm, or some equations might be added for better clarity.



**Questions:**

Is it possible to generate with Action level or Specific level only? It would be interesting to see the corresponding results, given there is a trend that the more diffusion steps for Specific level brings better performance.

The hierarchical nature of the method might be suitable for long sequence generation. Comparison on this would be an interesting thing to do.

For motion refinement, is it possible to modify the graph beyond changing edge weights, i.e., adding/deleting nodes? Or tuning some nodes' weight to zero. What would happen?



**Limitations:**

The author described some limitations of the methods, while it would be more helpful to discuss whether the current top-down paradigm could be extended in a bottom-up manner.

---

> ### Author Rebuttal · Authors · 2023-08-08
>
> We sincerely thank the reviewer for the insightful comments, and for noting that our method is "interesting". We address the questions as below.
>
> &nbsp;
>
> **Q1**: Is it possible to generate with action level or specific level only?
>
> **A1**: As suggested, we train networks to generate only the action level and specific level. As shown in the following table, the hierarchical diffusion model performs better on the FID metric with the same number of diffusion steps. We consider that it is because the hierarchical generation method can generate more detail than the single-stage generation method.
>
> | Methods             | Total steps |   R-Precision Top 1/2/3   |    FID    |  MM-Dist  | Diversity |
> | ------------------- | :---------: | :-----------------------: | :-------: | :-------: | :-------: |
> | GraphMotion         |     50      |   0.496/0.686/**0.778**   | **0.118** |   3.143   | **9.792** |
> | Action level only   |     50      | **0.507**/**0.692**/0.763 |   0.190   |   3.133   |   9.634   |
> | Specific level only |     50      |     0.492/0.686/0.772     |   0.144   | **3.090** |   9.710   |
>
> &nbsp;
>
> **Q2**: The hierarchical nature of the method might be suitable for long sequence generation.
>
> **A2**: Thanks for your insightful advice. We select motions with sequence frame lengths greater than 150 in the HumanML3D test set to evaluate the long sequence generation ability of the models. As shown in the following table, our method performs better on the FID metric than the baseline methods in generating long sequences. The results show that our method is significantly suitable for long sequence generation.
>
> | Methods            |     R-Precision Top 1/2/3     |    FID    |  MM-Dist  | Diversity | MModality |
> | ------------------ | :---------------------------: | :-------: | :-------: | :-------: | :-------: |
> | MDM (ICLR 2023)    |       0.295/0.467/0.572       |   0.594   |   5.497   |   8.994   | **2.874** |
> | MLD (CVPR 2023)    |       0.420/0.607/0.709       |   0.824   |   3.532   | **9.429** |   2.700   |
> | GraphMotion (Ours) | **0.442**/**0.644**/**0.744** | **0.260** | **3.411** |   9.167   |   2.702   |
>
> &nbsp;
>
> **Q3**: For motion refinement, is it possible to modify the graph beyond changing edge weights, i.e., adding/deleting nodes? Or tuning some nodes' weight to zero. What would happen?
>
> **A3**: Yes, we also perform the following operations on the hierarchical semantic graphs: **(1)** masking the node by replacing it with the MASK token; **(2)** modifying the node; **(3)** deleting nodes; **(4)** adding a new node. Please see Figure III in our global response for the details. The qualitative results demonstrate that our approach provides a novel method of refining generated motions.
>
> &nbsp;
>
> **Q4**: Qualitative comparison of different hierarchies is missed.
>
> **A4**: Thanks for your valuable suggestion. We provide different levels of qualitative comparison in the global response (Figure II). The results show that the output at the higher level (e.g., specific level) has more action details. Specifically, the motion level generates only coarse-grained overall motion. The action level generates local actions better than the motion level but lacks action specifics. The specific level generates more action specifics than the action level.
>
> &nbsp;
>
> **Q5**: The ablation on the design of the semantic graph is missed.
>
> **A5**: Thanks for your advice. We also tried another semantic graph based on the Stanford Scene Graph Parser. Unlike the proposed GraphMotion extracts three types (motions, actions, and specifics) of nodes, the Stanford Scene Graph Parser extracts triplet relationships (e.g., "woman"->"in"->"room"). We construct the semantic graph using the overall sentence as the first semantic level, the relation (e.g., "in") as the second semantic level, and the entities (e.g., "woman" and "room") as the third semantic level.
>
> As shown in the following table, the method based on Stanford Scene Graph Parser is not suitable for motion generation. We consider that it is because the triplet relation emphasizes the inter-entity relationship rather than actions, and the proposed GraphMotion builds the semantic graph with actions as the core, so GraphMotion is more suitable for motion generation.
>
>
> | Methods                                  | Total steps |     R-Precision Top 1/2/3     |    FID    |  MM-Dist  | Diversity |
> | ---------------------------------------- | :---------: | :---------------------------: | :-------: | :-------: | :-------: |
> | Stanford Scene Graph Parser based method |     150     |       0.497/0.683/0.781       |   0.212   |   3.105   | **9.789** |
> | GraphMotion                              |     150     | **0.504**/**0.699**/**0.785** | **0.116** | **3.070** |   9.692   |
>
> &nbsp;
>
> **Q6**: A more detailed description of the motion refining procedure, like a figure, an algorithm, or some equations might be added for better clarity.
>
> **A6**: Thanks for your suggestion. We have added detailed descriptions of the motion refining procedure in the revision. Specifically, for modifying the edge, we modify the attention coefficient $e$ as $e'=\gamma e+\delta$, where $\gamma$ and $ \delta$ are adjustable parameters. For modifying the node, we first extract text embedding through CLIP, and then replace the original node with this embedding or add it to the graph as a new node. Then the graph network updates all nodes. Finally, we use the updated nodes to generate new motions.
>
> &nbsp;
>
> **Q7**: It would be more helpful to discuss whether the current top-down paradigm could be extended in a bottom-up manner.
>
> **A7**: It is an interesting topic. A possible idea is the retrieval enhancement method, which generates motions based on action details retrieved from graph nodes.
>
> &nbsp;
>
> We sincerely thank you for your helpful comments. We will add the above important discussions in the final manuscript and highlight them. Thanks again for spending a huge amount of time on our paper.

---

> > ### Author Response · Authors · 2023-08-17
> >
> > Dear Reviewer
> >
> > Would it be possible for us to kindly ascertain if the provided responses have satisfactorily tackled any concerns you may have had and if further explanations or clarifications are needed? Your generous investment of time and effort in the evaluation of our work is truly commendable. We extend our heartfelt gratitude for your insightful commentary and the considerable time you have devoted to reviewing our paper.

---

> > ### Comment · Reviewer_E76p · 2023-08-19
> >
> > I appreciate the responses given by the authors. The newly conducted experiments have addressed my concerns. I would like to keep my positive rating.

---

> > > ### Author Response · Authors · 2023-08-19
> > > **Sincere appreciation**
> > >
> > > We sincerely thank you for your prompt and insightful review of our paper. Your comment is immensely appreciated and undoubtedly helps improve the quality of our work. We will add the above important discussions in the final manuscript and highlight them. Thanks again for taking the time and effort when handling our paper.

---

### Official Review · Reviewer_Xx1r · 2023-07-06

**Soundness:** 3 good
**Presentation:** 3 good
**Contribution:** 3 good
**Rating:** 6
**Confidence:** 4

**Summary:**

This work decomposes the motion description into three levels including motion, action and specifics, and proposes hierarchical semantic graphs to achieve fine-grained control of motion generation.
Experiments with the proposed method on HumanML3D and KIT datasets demonstrate better motion generation and more sensitive to subtle differences in texts than existing techniques.


**Strengths:**

The motivation is clear, and the proposed solutions are to the raised issues.
The results seem to be impressive and the experimental setup is somewhat reasonable.
The ability to continuously refine the generated motion is meaningful and helpful to the community.

**Weaknesses:**

1. The paper states several times that Transformer may overemphasize action names, but I think that the action level in the hierarchical semantic graph is also another kind of emphasis. Therefore, if possible, I hope the authors can experimentally demonstrate the "overemphasis" of the Transformer and prove that the hierarchical semantic graph does not have this problem.
2. Why is the last one in the action level of Figure 1 resumes instead of walking? How does it handle "squats to picks" and distinguish "walks and walking"? Also, the hierarchical semantic graph is built on a valid semantic parsing tool, and I'm not sure whether the tool will greatly affect the overall robustness of the model.
3. What are the results in the first row of Tables 3 and 4 in what configuration? By my understanding, these two rows should be the complete ablation of their respective parts, so they should be numerically different.
4. Is the modification in Figure 5 implemented in other latest works as well? Does it only modify the weights of the edges and can it modify the nodes?

**Questions:**

See the above weakness.

**Limitations:**

The authors fully explain the social impact of the work in the appendix, but only elaborate on the shortcomings of the used models (e.g., diffusion model and VAE) in terms of limitations, and do not well explain their own limitations and failure cases.

---

> ### Author Rebuttal · Authors · 2023-08-08
>
> Thanks for providing constructive feedback, and for noting that "the ability to continuously refine the generated motion is meaningful and helpful to the community." We address the questions below.
>
> &nbsp;
>
> **Q1**: Experimentally demonstrating the "overemphasis" of the Transformer and proving that the hierarchical semantic graph does not have this problem.
>
> **A1**: To demonstrate the "overemphasis" of the Transformer and prove that the hierarchical semantic graph does not have this problem, we mask the verbs and action names in the motion description to force the model to generate motion only from action specifics. For example, given the motion description "a person walks several steps forward in a straight line.", we would mask "walks".
>
> Transformer extracts text features automatically and implicitly. However, **it may encourage the model to take shortcuts, such as overemphasizing the action name "walks" at the expense of other important properties**. Therefore, when the verbs and action names are masked, the other models, which direct use the transformer to extract text features, fail to generate motion well.
>
> By contrast, the hierarchical semantic graph **explicitly extracts the action specifics**. The explicit factorization of the language embedding space facilitates a comprehensive understanding of motion description. It allows the model to **infer from action specifics such as "several steps forward" and "in a straight line" that the overall motion is "walking forward"**.
>
> As shown in the following table, our method can synthesize motion by relying only on action specifics, while other methods, which direct use the transformer, fail to generate motion well. These results indicate that the hierarchical semantic graph avoids the imbalance problem.
>
> | Methods            |    FID    |  MM-Dist  | Diversity | MModality |
> | ------------------ | :-------: | :-------: | :-------: | :-------: |
> | MDM (ICLR 2023)    |   5.622   |   7.163   |   8.713   |   3.578   |
> | MLD (CVPR 2023)    |   3.492   |   5.632   |   8.874   |   3.596   |
> | GraphMotion (Ours) | **1.826** | **5.530** | **9.284** | **3.699** |
>
> Besides, we also provide qualitative comparison in the global response (Figure I). When the verbs and action names are masked, the existing models, which direct use the transformer to extract text features, tend to generate motion randomly. In contrast, the generated results of our method better match the descriptions.
>
> &nbsp;
>
> **Q2**: Why is the last one in the action level of Figure 1 resumes instead of walking?
>
> **A2**: The last part of the original video is walking. But for ease of visualization, we sample 8 frames from the video and merge them into one figure. The walking part is not sampled and therefore fails to be shown. We are sorry for any confusion caused by this figure. In the revision, we have sampled more frames from the video to eliminate this confusion.
>
> &nbsp;
>
> **Q3**: How does it handle "squats to picks" and distinguish "walks and walking"?
>
> **A3**: In "squats to picks", both "squats" and "picks" are considered verbs and are therefore extracted as local actions. The time relationship between "squats" and "picks" is provided by the overall sentence (i.e., the global motion node). For "walks and walking", "walks" is considered a verb. For the semantic role parsing toolkit we use, "walking" might be considered action specific. For example, for "resumes walking", "walking" is taken as the action specific of the local action "resumes".
>
> &nbsp;
>
> **Q4**: The hierarchical semantic graph is built on a valid semantic parsing tool, and I'm not sure whether the tool will greatly affect the overall robustness of the model.
>
> **A4**: We explain the robustness of our method from two aspects. **First**, referring to the experiment in **A1**, our method can synthesize motion even after masking all verbs, which indicates our robustness. **Second**, semantic parsing tools are well-developed to handle simple texts in motion generation. Therefore, the robustness of our method is not a particular concern.
>
> &nbsp;
>
> **Q5**: What are the results in the first row of Tables 3 and 4 in what configuration?
>
> **A5**: Both the first row of Tables 3 and 4 represent the result of motion generation based on the vanilla Transformer and single-stage diffusion models. For comparison purposes, we present this baseline result in both Tables 3 and 4. We have added detailed descriptions in the revision to guide the readers.
>
> &nbsp;
>
> **Q6**: Is the modification in Figure 5 implemented in other latest works as well?
>
> **A6**: No, no other method can achieve this modification. To the best of our knowledge, we are the first to propose hierarchical semantic graphs, a fine-grained control signal, for text-to-motion generation. Besides, we are the first to propose modifying the edge weights of hierarchical semantic graphs to refine the generated results.
>
> &nbsp;
>
> **Q7**: Does it only modify the weights of the edges and can it modify the nodes?
>
> **A7**: It can modify the nodes. We provide additional qualitative analysis of refining motion results in the global response (Figure III). Specifically, we perform the following operations on the hierarchical semantic graphs: **(1)** masking the node by replacing it with the MASK token; **(2)** modifying the node; **(3)** deleting nodes; **(4)** adding a new node. The qualitative results demonstrate that our approach provides a novel method of refining generated motions.
>
> &nbsp;
>
> We sincerely thank you for your insightful comments. We will add the above important discussions in the final manuscript and highlight them. Thanks again for your great effort in improving our paper.

---

> > ### Author Response · Authors · 2023-08-17
> >
> > Dear Reviewer
> >
> > Could we kindly know if the responses have addressed your concerns and if further explanations or clarifications are needed? Your dedication to evaluating our work is deeply valued, and we are sincerely grateful for your perceptive comments and the substantial time you have dedicated to reviewing our paper.

---

> > ### Comment · Reviewer_Xx1r · 2023-08-18
> > **Some concerns that have not been addressed**
> >
> > Thank you for taking the time to comment. I am satisfied with some of the answers provided. However, there are still some concerns that have not been addressed: (1) Why are the first rows of Table 3 and Table 4 numerically the same? I think that they are ablation experiments done from different two aspects, so their baselines should not be the same. (2) Regarding limitations, the supplementary material only elaborates on the shortcomings of the used models (e.g., diffusion model and VAE) in terms of limitations, and does not well explain their own limitations and failure cases.

---

> > > ### Author Response · Authors · 2023-08-18
> > > **Further explanations and clarifications (1/2)**
> > >
> > > We sincerely thank you for your careful evaluation of our paper and for further discussion with us. We will provide further explanations and clarifications on the following questions.
> > >
> > > &nbsp;
> > >
> > > **Q1**: Why are the first rows of Table 3 and Table 4 numerically the same?
> > >
> > > **A1**: We apologize for any confusion caused by Tables 3 and 4. For further clarification, we provide the description of each row in Tables 3 and 4 in the following two tables. In fact, both the first row of Tables 3 and 4 represent the result of motion generation based on the vanilla Transformer and the single-stage diffusion model. We reiterate this result in both Tables 3 and 4 for the following reasons:
> > >
> > > * In Table 3, the first row acts as a baseline to measure the effectiveness of different components.
> > >
> > > * In Table 4, the first row is used to compare results at different levels to demonstrate that the intermediate result of our model is also better than the baseline. Especially because the structure of the motion level is very similar to the baseline, comparing the baseline results with the motion level results can further demonstrate the importance of semantic graph reasoning.
> > >
> > > We extend our sincere apologies for the oversight. To prevent any potential misunderstandings, we have incorporated comprehensive descriptions in the revised version. Furthermore, should you recommend the removal of the first row in Table 4, we are also open to its deletion to enhance the readability of the paper.
> > >
> > >
> > >
> > > | Rows in Table 3                                              | Description                                                  |
> > > | :----------------------------------------------------------- | :----------------------------------------------------------- |
> > > | **Row 1 (Baseline)**                                         | **Vanilla Transformer & Single-stage diffusion model**       |
> > > | Row 2 (+ Semantic Graph)                                     | All levels of the Hierarchical Semantic Graph & Single-stage diffusion model   |
> > > | Row 3 (+ Semantic Graph & Graph Reasoning)                   | All levels of the Hierarchical Semantic Graph & Graph Reasoning & Single-stage diffusion model |
> > > | Row 4 (+ Semantic Graph & Coarse-to-Fine Diffusion)          | All levels of the Hierarchical Semantic Graph & Three-stage diffusion model   |
> > > | Row 5 (+ Semantic Graph & Graph Reasoning & Coarse-to-Fine Diffusion) | All levels of the Hierarchical Semantic Graph & Graph Reasoning & Three-stage diffusion model (i.e., **the proposed GraphMotion**) |
> > >
> > > &nbsp;
> > >
> > > | Rows in Table 4                                      | Description                                                  |
> > > | :--------------------------------------------------- | :----------------------------------------------------------- |
> > > | **Row 1 (Baseline)**                                 | **Vanilla Transformer & Single-stage diffusion model**       |
> > > | Row 2 (Motion level)                                 | The first level of the Hierarchical Semantic Graph (with Graph Reasoning) & Single-stage diffusion model |
> > > | Row 3 (Motion level & Action level)                  | The first two levels of the Hierarchical Semantic Graph (with Graph Reasoning) & Two-stage diffusion model |
> > > | Row 4 (Motion level & Action level & Specific level) | All levels of the Hierarchical Semantic Graph (with Graph Reasoning) & Three-stage diffusion model (i.e., **the default setting for the proposed GraphMotion**) |

---

> > > ### Author Response · Authors · 2023-08-18
> > > **Further explanations and clarifications (2/2)**
> > >
> > > **Q2**: The supplementary material only elaborates on the shortcomings of the used models (e.g., diffusion model and VAE) in terms of limitations, and does not well explain their own limitations and failure cases.
> > >
> > > **A2**: Thanks for your valuable advice. We have conducted a more extensive analysis of the limitations of our method, as follows:
> > >
> > > * Our method can generate arbitrary length results but still under the max-length in the dataset. Modeling a continuous human motion with temporal consistency presents an intriguing aspect.
> > > * The current pipeline is limited to a singular form of motion representation. A more versatile pipeline that can seamlessly adapt to multiple datasets simultaneously would offer greater flexibility across diverse scenarios.
> > > * Since our method performs a diffusion process on the motion latent space rather than on the raw motion sequences, it is more suitable for high-level motion editing, such as style transfer, than for low-level motion editing, such as modifying the position of only one joint. Exploring low-level motion editing within latent space holds great promise and poses an exciting avenue for future research.
> > > * Our method inherits the randomness of diffusion models. While this characteristic contributes to the enhancement of diversity, it is important to acknowledge that it can occasionally lead to outcomes that are less desirable.
> > > * The human motion synthesis capabilities of our method are limited by the performance of the pre-trained motion variational autoencoders. Furthermore, delving into the realm of a more efficient motion latent space holds significant promise as a compelling avenue for future research.
> > > * Though our method brings negligible extra cost to computations, it is still limited by the slow inference speed of existing diffusion models. However, with the development of diffusion models, we anticipate a progressive mitigation of this limitation.
> > >
> > >
> > >
> > > We have added the above important discussions in the final manuscript. In addition, we will include some failure cases in the revision. There are two main reasons for these failure cases:
> > >
> > > * The randomness of diffusion models occasionally leads to outcomes that are less desirable, particularly noticeable in instances where longer sequences are generated, exacerbating the impact of randomness.
> > > * Due to its inadequate grasp of low-frequency words, the model struggles to generate motion that accurately aligns with the provided description containing many low-frequency words.
> > >
> > > &nbsp;
> > >
> > > We sincerely thank you for your valuable feedback. We will add the above important discussions in the final manuscript and highlight them. The dedication of time and effort you have devoted to a comprehensive review of our paper is genuinely appreciated.

---

### Official Review · Reviewer_xtyr · 2023-07-06

**Soundness:** 3 good
**Presentation:** 2 fair
**Contribution:** 3 good
**Rating:** 5
**Confidence:** 3

**Summary:**

This paper proposes a coarse-to-fine diffusion model coupled with a hierarchical semantic graphs to address the text-to-motion generation problem. To preserve the fine-grained control signals from captions, three-level textual features are extracted through GAT. Then, three diffusion models are adopted to recover the latent motion representations, which is decoded into motion sequences via VAE.

**Strengths:**

1. Semantic graphs are introduced to model the coarse-to-fine textual descriptions.
2. Hierarchical diffusion model is technically sound to learn the latent motion distributions.
3. The experimental results outperforms sota methods.
4. Fine-grained controllability of the proposed model is interesting.

**Weaknesses:**

1. The motivation behind the imbalance problem needs to be more convincing. In line 31-33, deficience of the existing method boils down to imbalance and coarseness. I am curious about if there are some examples could prove that imbalance problem exists in the other methods.
2. Many descriptions in this paper are misleading or incorrect, and it needs further explaination. In line 176, the term ``codebook size'' is ambiguous because it is generallly associated with VQVAE. In line 180-181, the purpose of diffusion process should be learning a mapping from gaussian noise sampled from $\mathcal{N}(0,1)$ to motion latent representation $z^m$, and the condition is the global motion node $V^m$. In line 183, $\beta_t$ is not step size, it is the noise schedule.

**Questions:**

Q1: Please present more examples to prove that imbalance problem exists in the other methods and how your method addresses it.

Q2: According to Table D and E in your supplementary material. R-Precision of your proposed VAE models outperforms Real motion, does this mean that R-Precision is not suitable for evaluating these two motion datasets?

**Limitations:**

The authors have discussed the potential negative societal impacts.

---

> ### Author Rebuttal · Authors · 2023-08-08
>
> Thanks for taking the time and effort when reading our paper and providing constructive comments. We address the questions below.
>
> &emsp;
>
> **Q1**: Please present more examples to prove that the imbalance problem exists in the other methods and how your method addresses it.
>
> **A1**: We will prove the imbalance problem of other methods from quantitative and qualitative perspectives respectively, and how our method addresses it.
>
> * From the quantitative perspective, we mask the verbs and action names in the motion description to force the model to generate motion only from action specifics. For example, given the motion description "a person walks several steps forward in a straight line.", we would mask "walks".
>
>   Transformer extracts text features automatically and implicitly. However, **it may encourage the model to take shortcuts, such as overemphasizing the action name "walks" at the expense of other important properties**. Therefore, when the verbs and action names are masked, the other models, which direct use the transformer to extract text features, fail to generate motion well.
>
>   By contrast, our method explicitly extracts the action specifics.  The explicit factorization of the language embedding space discourages the model from taking shortcuts. When the verbs and action names are masked, our method **can infer from action specifics such as "several steps forward" and "in a straight line" that the overall motion is "walking forward"**.
>
>   As shown in the following table, our method can synthesize motion by relying only on action specifics, while other methods fail to generate motion well. These results indicate that our method avoids the imbalance problem of other methods.
>
>   | Methods            |    FID    |  MM-Dist  | Diversity | MModality |
>   | ------------------ | :-------: | :-------: | :-------: | :-------: |
>   | MDM (ICLR 2023)    |   5.622   |   7.163   |   8.713   |   3.578   |
>   | MLD (CVPR 2023)    |   3.492   |   5.632   |   8.874   |   3.596   |
>   | GraphMotion (Ours) | **1.826** | **5.530** | **9.284** | **3.699** |
>
>
>
> * From the qualitative perspective, we provide qualitative comparison of the imbalance problem in the global response (Figure I). As shown in Figure I in the attached PDF, when the verbs and action names are masked, existing models tend to generate motion randomly. In contrast, our method can generate motion that matches the description based solely on the action specifics. These results show that our method is not overly focused on the verbs and action names.
>
> &emsp;
>
> **Q2**: According to Table D and E in your supplementary material. R-Precision of your proposed VAE models outperforms Real motion, does this mean that R-Precision is not suitable for evaluating these two motion datasets?
>
> **A2**: Since the existing methods are already very close to real motion in the R-Precision, we also believe that the R-Precision is not a good metric to evaluate these two motion datasets. For the motion generation task, it may be necessary to integrate a variety of metrics to evaluate the model performance.
>
> &emsp;
>
> **Q3**: Many descriptions in this paper are misleading or incorrect, and it needs further explanation.
>
> **A3**: Thank you for your helpful comments. We have corrected these descriptions in the revision. Specifically, we have made the following corrections:
>
> * We have amended "codebook size" to "token size" in line 176.
> * We have rewritten lines 180-181 to "our goal is to learn the diffusion process from Gaussian noise sampled  from $\mathcal{N}(0,1)$ to motion latent representation $z^m$, conditioned on the global motion node $\mathcal{V}^{m}$."
> * We have explained ${\beta}_t$ in line 183 by "${\beta}_t$ is the noise schedule."
>
> Besides, we have proofread our paper carefully in revision.
>
> &emsp;
>
> We sincerely thank you for your valuable comments. We will add the above important discussions in the final manuscript and highlight them. If you have further questions, please feel free to contact us. Thanks again for taking the time and effort on our paper.

---

> > ### Author Response · Authors · 2023-08-17
> >
> > Dear Reviewer
> >
> > Could we kindly inquire if the responses have satisfactorily tackled your concerns, or if there is a need for further clarification? Your commitment to reviewing our work is immensely appreciated, and we express our sincere gratitude for your insightful comments and the considerable time you have dedicated to reviewing our paper.

---

> > > ### Comment · Reviewer_xtyr · 2023-08-20
> > >
> > > The rebuttal has addressed my concerns, and I will keep my previous postive rating.

---

> > > > ### Author Response · Authors · 2023-08-20
> > > > **Sincere appreciation**
> > > >
> > > > We sincerely thank you for the dedication of time and effort you have devoted to a comprehensive review of our paper. We will add the above important discussions about the imbalance problem in the final manuscript and highlight them. Thanks again for taking the time and effort when handling our paper.

---

### Official Review · Reviewer_DRsM · 2023-07-07

**Soundness:** 3 good
**Presentation:** 3 good
**Contribution:** 3 good
**Rating:** 6
**Confidence:** 5

**Summary:**

This paper presents a novel motion generation pipeline that utilizes a 3-level hierarchical semantic graph. The entire reverse process of the motion diffusion model is divided into three stages: overall motion, local actions, and action specifics. The semantic graph is extracted through semantic role parsing and further enhanced using a Graph Attention Network. The obtained node features are subsequently input into a single transformer based on their semantic level.

**Strengths:**

1. The proposed GraphMotion approach demonstrates excellent performance in terms of numerical metrics on two datasets, showing significant improvements compared to existing methods. Particularly noteworthy is the detailed comparison of GraphMotion at different diffusion steps with other state-of-the-art methods, which adds more persuasive evidence to the results.

2. The motion refinement aspect is intriguing, as users are able to achieve a certain degree of motion editing by modifying the content of nodes. This adds an interesting and interactive element to the method.

3. The motivation and objectives of the entire approach are well-explained, making it easy to understand the rationale behind the proposed method.

**Weaknesses:**

1. It is recommended that the authors conduct user studies to quantitatively compare their method with other existing works based on qualitative results. The current demo video lacks sufficient comparisons, and additional comparisons should be included to provide a comprehensive evaluation. Furthermore, it would be beneficial to showcase the qualitative results of motion refinement in the demo video to provide a more complete demonstration of the method's capabilities.

**Questions:**

1. Why is the video length of MotionDiffuse shorter than other methods in the demo video?

2. Why is there a significant decrease in the FID metric on KIT-ML when the number of steps increases from 50 to 150?

3. Can GraphMotion perform low-level motion editing or motion composition similar to MotionDiffuse and MDM?

**Limitations:**

Limitations have been well discussed.

---

> ### Author Rebuttal · Authors · 2023-08-09
>
> We sincerely thank the reviewer for the constructive comments, and for noting that our method is "novel", "intriguing" and "well-explained". We address the questions as below.
>
> &nbsp;
>
> **Q1**: Why is the video length of MotionDiffuse shorter than other methods in the demo video?
>
> **A1**: This is because we refer to the settings provided by the repository of MotionDiffuse. The repository of MotionDiffuse recommends setting the generated motion sequence length to 60 frames, so we follow this setting when visualizing, which makes the generated video short.
>
> &nbsp;
>
> **Q2**: Why is there a significant decrease in the FID metric on KIT-ML when the number of steps increases from 50 to 150?
>
>  **A2**: We explain this phenomenon from two perspectives:
>
> * Due to the relatively small scale of the KIT-ML dataset, motion sequences can be generated well when the number of diffusion steps is 50.
> * Since we use DDIM instead of DDPM, increasing the number of diffusion steps does not introduce additional randomness and may reduce the diversity of the generated results. We find that when the number of DDIM diffusion steps increase, even though the fluency of generated motions increases, the diversity of generated motions reduces, leading to a decrease in the FID metric.
>
> Therefore, to balance generation quality and generation diversity, setting the number of steps to 50 on the KIT-ML dataset is a better choice than setting the number of steps to 150.
>
> &nbsp;
>
> **Q3**: Can GraphMotion perform low-level motion editing or motion composition similar to MotionDiffuse and MDM?
>
> **A3**: GraphMotion can perform in-betweening motion editing. Unlike MotionDiffuse and MDM, which implement motion composition during the reverse diffusion process, GraphMotion can use the motion decoder $\mathcal{D}$ to fill new motion frames from the motion latent space and fixed motion frames.
>
> In addition, since GraphMotion performs a diffusion process on the motion latent space rather than on the raw motion sequences, GraphMotion is more suitable for high-level motion editing, such as style transfer. Specifically, the input motion is first encoded into the motion latent space by the motion encoder $\mathcal{E}$. Then we can use the energy-guided diffusion process [1] to make overall motion editing according to the text.
>
> [1] Yu, Jiwen, et al. "Freedom: Training-free energy-guided conditional diffusion model." *arXiv preprint arXiv:2303.09833* (2023).
>
> &nbsp;
>
> **Q4**: It is recommended that the authors conduct user studies for quantitative comparison.
>
> **A4**: Thanks for your valuable suggestion. We randomly selected 39 motion descriptions for the user study. The results are shown in the following table. Each row represents the preference rate of GraphMotion over the compared model. GraphMotion is preferred over the other models most of the time.
>
> | **Methods**                   | Preference Rate of GraphMotion|
> | ----------------------------- | :-------------: |
> | GraphMotion vs. MotionDiffuse |     64.10%      |
> | GraphMotion vs. MLD           |     56.41%      |
> | GraphMotion vs. Ground Truth  |     48.72%      |
>
> &nbsp;
>
> **Q5**: It would be beneficial to showcase the qualitative results of motion refinement in the demo video to provide a more complete demonstration of the method's capabilities.
>
> **A5**: Thanks for your advice. We will showcase the qualitative results of motion refinement in the demo video, such as modifying the edge. In addition, we will also showcase the qualitative results of new operations in the global response (Figure III), i.e., **(1)** masking the node by replacing it with the MASK token; **(2)** modifying the node; **(3)** deleting nodes; **(4)** adding a new node.
>
> &nbsp;
>
> We sincerely thank you for your constructive comments. We will add the above important discussions in the final manuscript and highlight them. Thanks again for taking the time and effort on our paper.

---

> > ### Author Response · Authors · 2023-08-17
> >
> > Dear Reviewer
> >
> > May we kindly inquire if the provided responses have adequately addressed any questions you might have had? If there remains a requirement for further explanations or clarifications? We wish to express our sincere gratitude for your meticulous evaluation and for generously investing a significant amount of your time in reviewing our paper. Your feedback would be greatly valued.

---

> ### Comment · Reviewer_DRsM · 2023-08-17
> **Post-rebuttal Comment**
>
> The rebuttal addressed most of my concerns and I will keep the postive view on the paper.

---

> > ### Author Response · Authors · 2023-08-18
> > **Sincere appreciation**
> >
> > Thank you for your invaluable feedback. Your expertise and attention to detail have been invaluable in guiding us toward a clearer presentation. The dedication of time and effort you have devoted to a comprehensive review of our paper is genuinely appreciated.

---

### Author Rebuttal · Authors · 2023-08-08

# Global Response

We sincerely thank all PCs, SACs, ACs, and Reviewers for their time and efforts when handling our paper.

&nbsp;

All reviewers appreciate the contributions of our method:

* Both Reviewers DRsM and Xx1r point out that "**the motivation is clear**, and **the proposed solutions are to the raised issues**."
* Reviewers DRsM, Xx1r, and E76p all comment that "**the experimental setup is somewhat reasonable**" and "**the detailed comparison** of GraphMotion at different diffusion steps with other state-of-the-art methods, which adds **more persuasive evidence to the results**."
* Reviewers DRsM, xtyr, Xx1r, and E76p all mention that "the proposed GraphMotion approach demonstrates **excellent performance** in terms of numerical metrics on two datasets, showing **significant improvements** compared to existing methods."
* Reviewers DRsM, xtyr, Xx1r, and E76p all point out that the motion refinement aspect is "**intriguing**", and "**meaningful and helpful to the community**".

&nbsp;

As suggested by the reviewers, we provide the following in the attached PDF:

* **Figure I: Qualitative comparison between our method and other methods on the imbalance problem.** To demonstrate the imbalance problem of other methods and prove that our method does not have this problem, we mask the verbs and action names in the motion description to force the model to generate motion only from action specifics. When the verbs and action names are masked, existing models tend to generate motion randomly. In contrast, our method can generate motion that matches the description based solely on the action specifics. These results show that our method is not overly focused on the verbs and action names.
* **Figure II: Qualitative comparison of different hierarchies.** The results show that the output at the higher level (e.g., specific level) has more action details. Specifically, the motion level generates only coarse-grained overall motion. The action level generates local actions better than the motion level but lacks action specifics. The specific level generates more action specifics than the action level.
* **Figure III: Additional qualitative analysis of refining motion results.** We perform the following operations on the hierarchical semantic graphs: **(1)** masking the node by replacing it with the MASK token; **(2)** modifying the node; **(3)** deleting nodes; **(4)** adding a new node. The qualitative results demonstrate that our approach provides a novel method of refining generated motions, which may be meaningful and helpful to the community.

&nbsp;

We will add the above important qualitative analysis in the final manuscript and highlight them. Thanks again for taking the time and effort on our paper.

---

### Decision · Program_Chairs · 2023-09-21

**Decision:**

Accept (poster)

**Comment:**

This paper was reviewed by four experts in the field. Based on the reviewers' feedback, the decision is to recommend the paper for acceptance to NeurIPS 2023. The reviewers did raise some valuable concerns that should be addressed in the final camera-ready version of the paper. The authors are encouraged to make the necessary changes to the best of their ability. We congratulate the authors on the acceptance of their paper!